# Dynamic curvature regulation accounts for the symmetric and asymmetric beats of *Chlamydomonas* flagella

Pablo Sartori[1†], Veikko F Geyer[2†], Andre Scholich[1], Frank Jülicher[1], Jonathon Howard[2*]

[1]Max Planck Institute for the Physics of Complex Systems, Dresden, Germany; [2]Department of Molecular Biophysics and Biochemistry, Yale University, New Haven, United States

**Abstract** Cilia and flagella are model systems for studying how mechanical forces control morphology. The periodic bending motion of cilia and flagella is thought to arise from mechanical feedback: dynein motors generate sliding forces that bend the flagellum, and bending leads to deformations and stresses, which feed back and regulate the motors. Three alternative feedback mechanisms have been proposed: regulation by the sliding forces, regulation by the curvature of the flagellum, and regulation by the normal forces that deform the cross-section of the flagellum. In this work, we combined theoretical and experimental approaches to show that the curvature control mechanism is the one that accords best with the bending waveforms of *Chlamydomonas* flagella. We make the surprising prediction that the motors respond to the time derivative of curvature, rather than curvature itself, hinting at an adaptation mechanism controlling the flagellar beat.

**\*For correspondence:** jonathon. howard@yale.edu

[†]These authors contributed equally to this work

## Introduction

Cilia and flagella are long, thin organelles whose oscillatory bending waves propel cells through fluids and drive fluid flows across the surfaces of cells. The internal motile structure, the axoneme, contains nine doublet microtubules, a central pair of single microtubules, motor proteins of the axonemal dynein family and a large number of additional structural and regulatory proteins (*Pazour et al., 2005*). The axonemal dyneins power the beat by generating sliding forces between adjacent doublets (*Summers and Gibbons, 1971*; *Brokaw, 1989*). The sliding is then converted to bending (*Satir, 1965*) by constraints at the base of the axoneme (e.g. the basal body) and/or along the length of the axoneme (e.g. nexin links) (*Brokaw, 2009*).

While the mechanism by which the sliding of a doublet is converted to bending is well established, it is not known how the activities of the dyneins are coordinated in space and time to produce the periodic beating pattern of the axoneme. For example, bending the axoneme in one direction requires higher dynein activity on one side of the axoneme than on the other; if the activities are equal then the forces will cancel and there will be no bending. Therefore, to alternatively bend in one direction and then the other requires dynein activity to alternate between the two sides (*Satir and Matsuoka, 1989*). The switching of dynein activity is rapid, taking place twice per beat cycle and at rates above 100 times per second for *Chlamydomonas*. The coordination required for such rapidly alternating bending is thought to result from mechanical feedback: the axonemal dyneins generate forces that bend and deform the axoneme; and the deformations, in turn, regulate the dyneins. Because of the geometry of the axoneme, deformation leads to stresses and strains

that have components in various directions (e.g. axial and radial). However, which component (or components) regulates the dyneins is not known.

Three different, but not mutually exclusive, molecular mechanisms for dynein coordination have been suggested in the literature (*Figure 1D*). (i) In the sliding control-mechanism, the dyneins behave as 'slip bonds': they detach when subject to forces acting parallel to the long axis of the microtubule doublets and that oppose sliding (*Brokaw, 1975*; *Jülicher and Prost, 1997*; *Camalet and Jülicher, 2000*; *Riedel-Kruse et al., 2007*). The build-up of sliding forces on one side of the axoneme therefore induces detachment of the dyneins on the other side (and vice versa): the two sides are antagonistic. The resolution of this reciprocal inhibition or 'tug of war' is a catastrophic detachment of dyneins on one side of the axoneme, leading to an imbalance of sliding forces and therefore to axonemal bending. (ii) In the curvature-control mechanism, the detachment of dynein is regulated by doublet curvature (*Morita and Shingyoji, 2004*; *Brokaw, 1972*; *Brokaw, 2009*). This leads to a similar reciprocal inhibition because the sign of the curvature is opposite on opposite sides of the axoneme. (iii) In the normal-force control mechanism, also called the geometric clutch, the detachment of dynein is regulated by transverse forces that act to separate adjacent doublets when they are curved (*Lindemann, 1994b*). Which, if any, of these mechanisms regulates the beating of the axoneme is not known.

In this work, we developed a two-dimensional mathematical model of the axoneme that can incorporate any or all of these different feedback mechanisms. The model extends earlier models (*Camalet and Jülicher, 2000*; *Riedel-Kruse et al., 2007*) by including static curvature, which gives rise to asymmetric beats such as those of *Chlamydomonas* and of ciliated epithelial cells. This model is similar to a recent model (*Bayly and Wilson, 2014*). We then tested the different feedback mechanisms by comparing the predictions of the associated models with high spatial and temporal resolution measurements of the bending waveforms of isolated, reactivated *Chlamydomonas* axonemes (*Hyams and Borisy, 1975*; *Bessen et al., 1980*). We found that the curvature-control mechanism accorded with experiments using both wild type cells, which have an asymmetric beat, and with experiments using the *mbo2* mutant, which has a nearly symmetric beat. By contrast, the sliding mechanism gave poor fits to both the wild type and *mbo2* data, and the normal-force mechanism gave unsatisfactory fits to the *mbo2* data.

## Theoretical model

### Two-dimensional model of the axoneme

In the two dimensional model, we project the cross-section of the three-dimensional axoneme onto a pair of filaments (*Figure 1A,B*). The projection retains the key idea that motors on opposite sides of the axoneme generate bends in opposite directions (*Satir and Matsuoka, 1989*). The dyneins that give rise to the principle bend  (this corresponds to bends that lie on the outside of the curved path along which an axoneme swims [*Gibbons and Gibbons, 1972*]; *Figure 1A,B* green), which in *Chlamyodomonas* has the same sign as the static curvature, are combined to generate sliding in one direction. The dyneins that give rise to the reverse bend (*Figure 1A,B* blue), are combined to generate sliding in the opposite direction.

The two filaments have the same polarity (all motors move towards the base) and have combined bending rigidity $\kappa$. They are assumed to be inextensible and held together by elastic elements that maintain the filaments at a constant distance ($a$) from each other. Although there is some evidence that doublet separation is not constant (*Lindemann and Mitchell, 2007*), our assumption simplifies the theory while still allowing us to calculate the normal force in the elastic elements that keeps the separation constant (*Mukundan et al., 2014*). In our normal-force mechanism the motors are regulated by the force rather than by the separation of the doublets, as in *Lindemann (1994a)*. However, if the normal force is small, the separation is proportional to the force and the two models are equivalent.

The filaments are immersed in an aqueous fluid and experience drag forces arising from the fluid viscosity. The hydrodynamic forces are proportional to the velocity, with friction coefficients $\xi_\mathrm{n}$ and $\xi_\mathrm{t}$ (per unit length) for motion normal to and tangential to the axis of the filament. The values of the mechanical parameters are estimated in *Appendix 5*.

The position of each point on the filament pair is specified at each time by the vector $\mathbf{r}(s)$, a function of the arc-length $s$, along the centerline between the filaments. Calculating the tangent vector

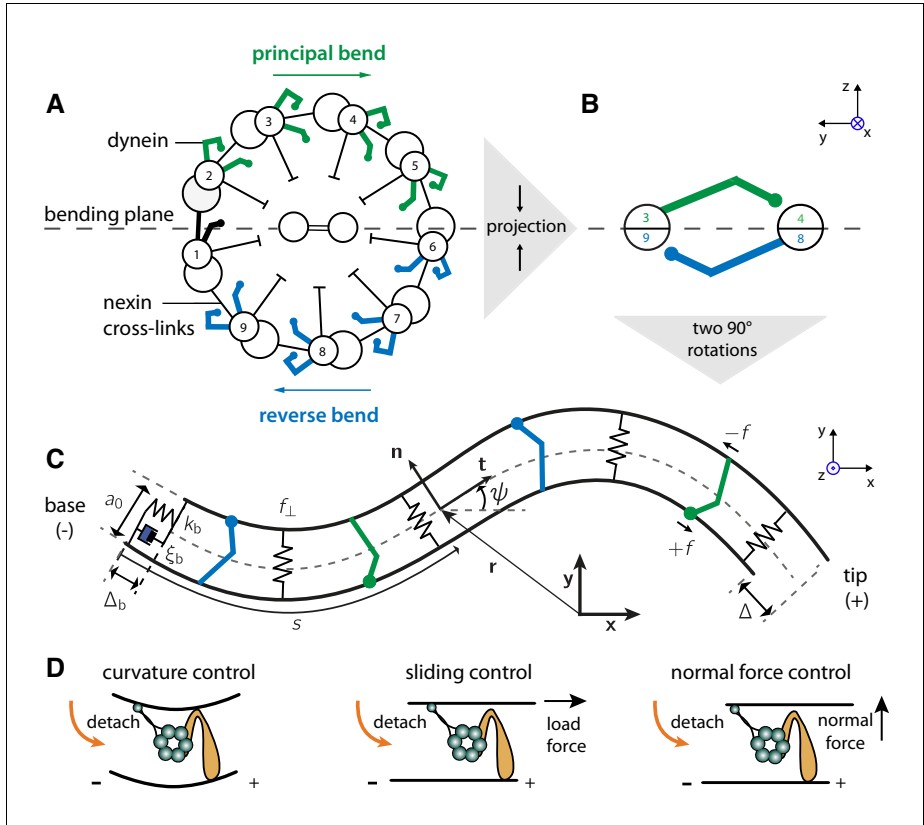

**Figure 1.** Two-dimensional model of the axoneme and mechanisms of dynein regulation. (**A**) Cross-section of an axoneme, as seen from the basal end looking towards the distal tip. The numbering of the doublets follows the convenstion for *Chlamydomonas* (**Hoops and Witman, 1983**). The green dyneins bend the axoneme such that the center of curvature is to the right. (**B**) The projection of the cross section onto two filaments to form the two-dimensional model. The green and blue motors bend the filament pair in opposite directions. (**C**) Two-dimensional model of the axoneme, as seen in the bending plane, **xy**. The two filaments are constrained to have spacing $a$. The point at arc-length $s$ has position vector **r** (relative to the origin), tangent vector **t**, normal vector **n**, and tangent angle $\psi$ with respect to the horizontal axis of the lab-frame **xy**. Dyneins on the upper filament (green) have microtubule-binding domains (MTBs, denoted by filled circles) that walk along the lower filament towards the base and produce a (tensile) force density $+f$ on the lower filament. This force slides the lower filament towards the distal end. The dyneins on the opposite filament (blue) create sliding (and bending) forces in the opposite direction. The local sliding displacement is given by $\Delta$, and the sliding at the base is $\Delta_b$. The sign convention is defined in *Figure 10* and *Appendix 4*. The springs between the filaments oppose filament separation by the normal force, $f_\perp$. The spring and dashpot at the base consitute the basal compliance, with stiffness $k_b$ and friction coefficient $\xi_b$. (**D**) Schematic of dynein regulation mechanisms. In *curvature control* the dynein MTB detaches due to an increase in curvature. In *sliding control* detachment is enhanced by a tangential loading force, and in *normal force control* the normal force, which tends to separate the filaments, enhances detachment. Signs indicate doublet polarity.

as $\mathbf{t}(s) = \dot{\mathbf{r}}(s)$, where dots denote arc-length derivatives, allows us to define the tangent angle, $\psi(s)$, with respect to the horizontal axis of the laboratory frame. The tangent angle characterizes the shape of the filament. For a given filament shape, the pair of filaments will have a local sliding displacement, $\Delta(s)$. We assume that the filaments are incompressible, though they can support tension along their centerline. For incompressible filaments, sliding is linearly related to the tangent angle via

$$\Delta(s) = \Delta_b + a[\psi(s) - \psi(0)], \tag{1}$$

where $\Delta_b = \Delta(0)$ is the basal sliding (*Figure 1C*). The sign convention is defined in *Figure 10* and *Appendix 4*.

To compare the observed periodic beats to those predicted by the model, we write the tangent angle as a sum of Fourier modes

$$\psi(s,t) = \sum_{n=-\infty}^{\infty} \psi_n(s)\exp(in\omega t), \tag{2}$$

where $\omega$ is the angular frequency of the beat ($\omega/2\pi$ is the beat frequency in cycles per second), $t$ is time, $i$ is the imaginary unit, and $\psi_n$ are the modes (indexed by $n$). The modes are complex-valued functions of arc-length that represent the amplitude and phase of the beat waveforms. They satisfy $\psi_{-n} = \psi_n^*$ to keep the angle real (* represents the complex conjugate). For each value of $n>1$, there is an associated equation of motion. We refer to $n=0$ as the static mode, which corresponds to the time-averaged shape. We refer to $n=1$ as the fundamental mode, which corresponds to the dynamic shape at the beat frequency. The static and fundamental modes are the only ones considered in this work. In this case, we can write $\psi(s,t) = \psi_0(s) + |\psi_1(s)| \, sin[\omega t + \phi(s)]$ where $|\psi_1(s)|$ is the amplitude of the fundamental mode and $\phi(s)$ is the phase ($arg\psi$). We use the complex formulation because it simplifies the theory, whose goal is to make predictions for $|\psi_1(s)|$ and $\phi(s)$ to compare to the experimental data (e.g. *Figure 3ii–iii*). The same modal decomposition can be done for all other parameters that depend on time including the sliding displacement, $\Delta(s)$, the basal sliding, $\Delta_b$, the sliding force between the filaments, $f$, and the tension, $\tau$, in the centerline of the filaments.

## Force balance in the axoneme

The shape of the axoneme depends on the balance between the mechanical forces (from the motors and the elastic elements) and the hydrodynamic forces. The mechanical force is the derivative (with respect to the position vector **r**) of the work functional $U$ (*Appendix 1*, *Equation 11*), which depends on the bending rigidity of the doublets, the stiffness of the cross linkers, motor forces and the shape of the filament. This force is balanced by the hydrodynamic force from the fluid, which is proportional to the velocity ($\partial_t\mathbf{r}$) of the axoneme at each point: $-\hat{\Pi} \cdot \partial_t\mathbf{r} = \delta U/\delta\mathbf{r}$, where $\hat{\Pi}$ is the friction matrix (*Lauga and Powers, 2009*). For a slender body at low Reynolds number the friction matrix is $\hat{\Pi} = \xi_n\mathbf{nn} + \xi_t\mathbf{tt}$, where $\mathbf{n}(s)$ a unit vector normal to the tangent vector $\mathbf{t}(s)$ (*Figure 1C*). Force balance yields non-linear equations of motion (*Equations 14–16* in *Appendix 1*, also see *Sartori, 2015*). We then derive the static solution, corresponding to mode $n=0$, and the periodic solution at the beat frequency, corresponding to mode $n=1$.

## Static mode

If the motor forces do not change with time, then the axoneme will not move and the hydrodynamic forces are zero, causing the tension in the axoneme to be zero. The static mode of the filament pair can be calculated from the static force-balance equation

$$\kappa\dot{\psi}_0(s) = aF_0(s), \qquad F(s) = -\int_s^L f(s')\mathrm{d}s', \tag{3}$$

where $\dot{\psi}_0(s)$, the arc-length derivative of the tangle angle, is the curvature of the static mode and $F_0(s)$ is the static mode of the integrated motor force (*Appendix 1*, *Equation 19*). The static mode is the time-averaged shape which, in *Chlamydomonas* axonemes, has approximately constant curvature all along the length, and as a result $\dot{\psi}_0(s) \approx C_0$ (*Geyer et al., 2016*; *Eshel and Brokaw, 1987*) (see Results). In our theoretical analysis, we therefore ignore deviations from constant curvature. Such deviations are not expected to significantly affect the conclusions of this work because we found that the static curvature had little effect on the dynamics (the wild type and *mbo2* axonemes have similar beats, see Results). According to the static force-balance equation, bending an axoneme into a shape with constant curvature requires the integrated motor force to be independent of arc-length, which in turn requires the motor forces be concentrated near the distal end (*Mukundan et al., 2014*). We therefore approximate the static component of the motor force density by

$$f_0 = -\delta(s - L)\kappa C_0/a, \tag{4}$$

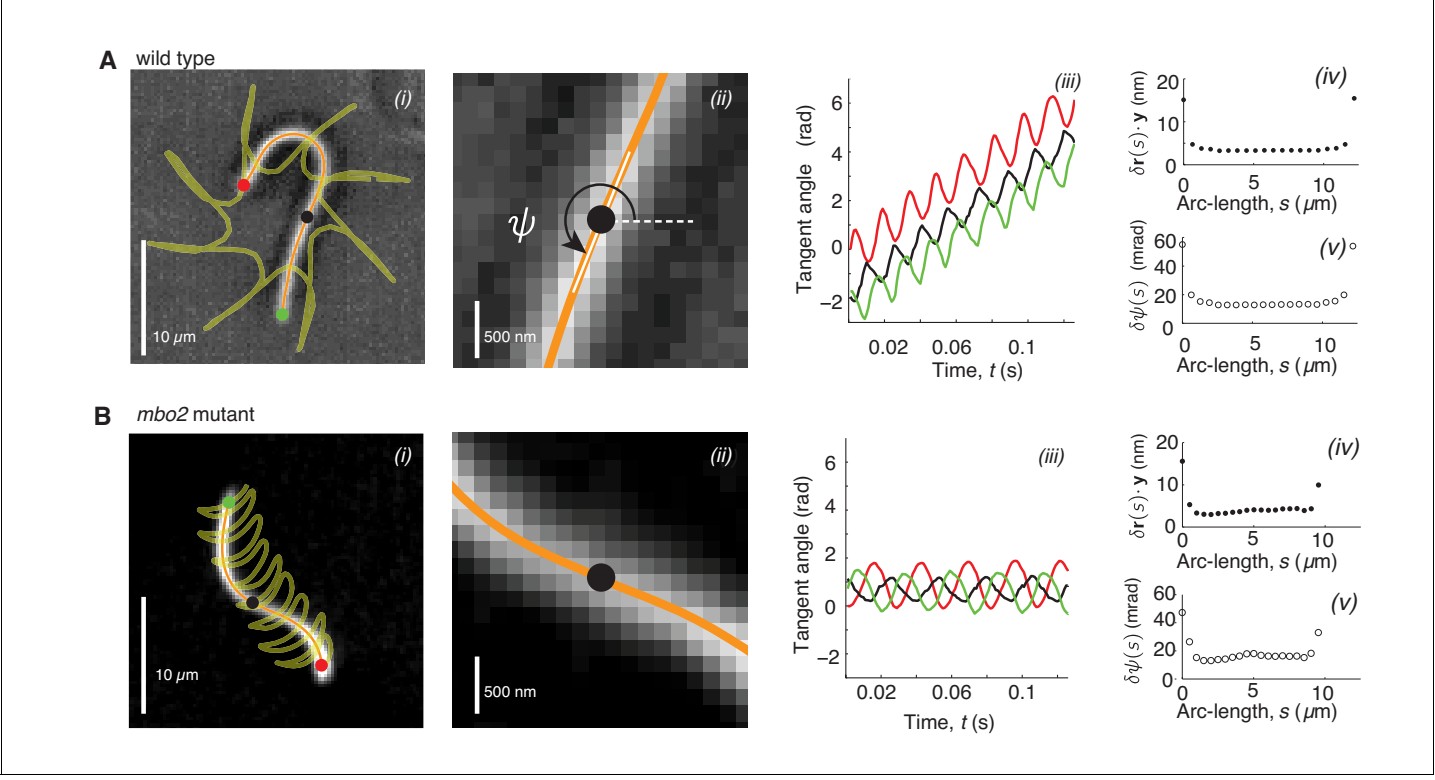

**Figure 2.** High precision tracking of isolated, reactivated axonemes. Panel A corresponds to *Chlamydomonas* wild type axonenes, panel B to *mbo2* mutant axonemes. (**A**) (*i*) Inverted phase-contrast image of a wild type axoneme. The orange curve represents the tracked centerline. The points depict the basal end (red), the distal end (green) and the center position (black) of the axoneme. The yellow line depicts the trajectory of the basal end, which is the leading end during swimming (*Geyer, 2012*). (*ii*) Same image as in A*i*, magnified around the center region. The tangent angle $\psi(s,t)$ is defined with respect to the lab frame. (*iii*) Tangent angle at three different arc-length positions (depicted in A*i*) as a function of time. The linearly increasing tangent angle corresponds to a counter-clockwise rotation of the axoneme during swimming. Mean uncertainty measured over 1000 adjacent frames of the position $\delta\mathbf{r} \cdot \mathbf{y}$ (*iv*) (the $\delta\mathbf{r} \cdot \mathbf{x}$ error gives a similar result) and the tangent angle $\delta\psi$ in (*v*). (**B**) is analogous to (**A**), but for *mbo2* mutant axonemes.

where $\delta$ is the Dirac delta function, and the minus sign ensures that a positive dynein force produces a negative curvature in accord with our sign convention (*Appendix 4*).

## Dynamic mode

To obtain the equation of motion at the beat frequency (i.e the $n = 1$ dynamic mode), we substitute the modal expansions of the tangent angle (*Equation 2*), the motor force and the tension into the non-linear dynamic equations and keep only the terms at the fundamental frequency ($n = 1$):

$$i\omega\xi_n\psi_1 = -\kappa\ddddot{\psi}_1 + a\ddot{f}_1 + [1 + \xi_n/\xi_t]C_0\dot{\tau}_1 + (\xi_n/\xi_t)C_0^2(\kappa\ddot{\psi}_1 - af_1)$$
$$(\xi_n/\xi_t)\ddot{\tau}_1 - C_0^2\tau_1 = -[1 + (\xi_n/\xi_t)]C_0(\kappa\dddot{\psi}_1 - a\dot{f}_1)$$

(5)

(*Appendix 1*, *Equations 14–16*, *Appendix 2*, *Equation 21*). Associated with these equations are boundary conditions specifying the tangent angle (and its spatial derivatives) and the tension at the basal and distal ends (*Appendix 2*, *Equation 20*). The boundary conditions for a freely swimming axoneme correspond to no external forces or torques acting at the ends.

*Equation 5* generalizes previous models of symmetrically beating axonemes (*Machin, 1958*; *Riedel-Kruse et al., 2007*; *Camalet and Jülicher, 2000*, in which the static curvature ($C_0$) and the axial tension ($\tau_1$) are zero. For $C_0 \neq 0$ new terms appear, and the system of equations is of order six rather than four in the symmetric case. The magnitude of the new terms can be estimated by considering the plane-wave approximation, in which $\psi_1(s) = \exp(-2\pi is/\lambda)$, with $\lambda \approx L$ the wavelength. The plane wave is sinusoidal, $\psi(s,t) = \sin(\omega t - 2\pi s/\lambda)$. Though the plane wave is only an approximation to the

shape and does not satisfy the boundary conditions (e.g. the curvature at the distal end of an axoneme is always zero), it is nevertheless useful for calculating approximate values of parameters. For example, the fourth term on the right hand side of the upper equation is of order $(\xi_{\mathrm{n}}/\xi_{\mathrm{t}})C_0^2\kappa(2\pi/\lambda)^2$ and is in phase with the first term, which is of order $\kappa(2\pi/\lambda)^4$. For *Chlamydomonas* axonemes $\lambda \sim L$ and $C_0 \sim \pi/L$, and since $\xi_{\mathrm{n}}/\xi_{\mathrm{t}} \approx 2$ (*Appendix 5*) the ratio of these terms is $\sim 0.5$. A similar reasoning shows that the third term is in anti-phase, and its contribution is of order $\sim 1$. This shows that the new terms, which enter through the asymmetry, can not be neglected *a priori*. Thus, for the large observed asymmetry of the *Chlamydomonas* axoneme, we expect that there is coupling between the $n=0$ and $n=1$ modes, significantly modifying the dynamics of the beating axoneme. As we will see, the static curvature $C_0$ has little effect in the curvature-control model but has a large effect in the normal-force model.

*Equation 5* shows how an oscillatory active sliding force, $f(s,t) = f_1(s)e^{i\omega t} + f_1^*(s)e^{-i\omega t}$, can produce dynamic bending of the axoneme (with appropriate parameter and boundary conditions). To see this, note that the upper equation can be rearranged to provide an expression for $\tau_1$ in terms of $\psi_1$ and $f_1$ (and their derivatives). This expression can then be substituted into the lower equation, to provide a relationship between $\psi_1$ and $f_1$, so that if $f_1$ is known, $\psi_1$ can be calculated. A trivial example is when $f_1 = 0$, in which case the only solutions are $\psi_1 = 0$ and $\tau_1 = 0$. If $f_1$ is non zero, the equation may have non-trivial solutions, corresponding to bending oscillations.

## Three mechanisms of motor control

We now build the three molecular mechanisms of motor control into the two-dimensional model of the axoneme. *Equation 5* shows that an oscillating sliding force can produce a dynamic beating pattern. However, we do not expect the motor proteins themselves to be the oscillators, because oscillations have never been observed in single-molecule recordings (but see *Shingyoji et al., 1998*). Rather, we expect that the sliding forces generated by the dyneins are regulated directly or indirectly by the shape of the axoneme. Such regulation constitutes mechanical feedback. If we have an expression for how the motor force depends on the tangent angle (and its derivatives), this can be substituted into the equation of motion, which can then be solved (using the boundary conditions) to predict bending waveforms.

The most general linear expression for the dependence of the motor force on sliding, curvature and normal force is

$$f_n(s) = \chi(n\omega)\Delta_n(s) + \beta(n\omega)\dot{\psi}_n(s) + \gamma(n\omega)f_{\perp,n}(s), \tag{6}$$

where $n \geq 0$ is the mode index. $\chi(\omega)$, $\beta(\omega)$ and $\gamma(\omega)$ are complex, frequency-dependent coefficients describing how the motor force responds to sliding, curvature and normal forces, respectively. The coefficients depend on the molecular properties of the dyneins such as their density along the doublets, their force-velocity curves, and the sensitivity of their unbinding on load force, on curvature or on normal force; they also depend on the elastic and viscous resistance to sliding between the doublets (*Camalet and Jülicher, 2000*; *Riedel-Kruse et al., 2007*). If the density of motors and the mechanical properties are independent of arc-length, then the coefficients will also be independent of the arc-length. Though light and electron microscopy studies have shown that there are longitudinal variations in dynein isoforms and dynein density (*Yagi et al., 2009*; *Bui et al., 2012*), we ignore these variations in the present work. Such variation could be included in more elaborate models (e.g. *Riedel-Kruse et al., 2007*). The coefficients are complex because in general the force could depend on the instantaneous value of the parameter (the real part) and or on the rate of change of the parameter (the imaginary part). To see this, suppose that the force depends on the sliding displacement ($\Delta$) and the sliding velocity ($\partial_t\Delta$). Then $f(s,t) = k\Delta(s,t) + \xi\partial_t\Delta(s,t)$, and in the Fourier representation $f_1(s) = (k + i\omega\xi)\Delta_1(s)$, so $\chi = k + i\omega\xi$. Finally, the coefficients will in general depend on frequency through delays caused by the finite detachment times. The response coefficients are the generalization to active motors of the linear response coefficient of a passive system. For example, the basal force of the axoneme, $F_{\mathrm{b}} = F(0)$, is described by $F_{\mathrm{b}} = k_{\mathrm{b}}\Delta_{\mathrm{b}} + \xi_{\mathrm{b}}\partial_t\Delta_{\mathrm{b}}$, where $k_{\mathrm{b}}$ is the basal stiffness and $\xi_{\mathrm{b}}$ is the basal damping coefficient. Thus, the fundamental mode of the basal force is given by $F_{\mathrm{b},1} = \chi_{\mathrm{b}}(\omega)\Delta_{\mathrm{b},1}$ with basal impedance $\chi_{\mathrm{b}}(\omega) = k_{\mathrm{b}} + i\omega\xi_{\mathrm{b}}$.

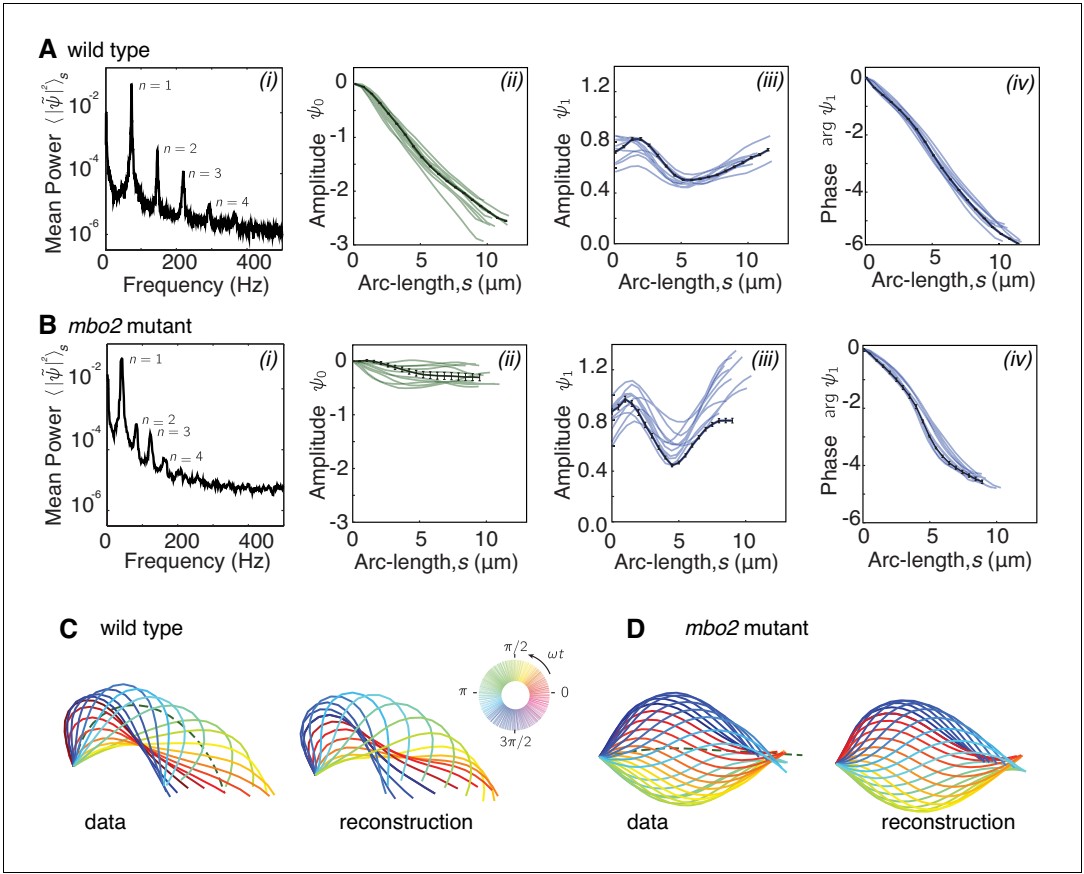

**Figure 3.** Fourier decomposition of the beat. Panel A shows the Fourier decomposition of the waveform of wild type axonemes, panel B the decomposition of the *mbo2* axoneme waveform. (A) (*i*) Power spectrum of the tangent angle averaged over arc-length. The fundamental mode ($n = 1$) and three higher harmonics ($n = 2, 3, 4$) are labeled. (*ii*) Angular representation of the static ($n = 0$) mode as a function of arc-length. The approximately constant slope indicates that the static curvature is close to constant $\dot{\psi}_0 = C_0$. (*iii*) The amplitude and phase (argument) of the fundamental mode are shown in *iii* and *iv*, respectively. The approximately linear decrease in phase indicates steady wave propagation. The data of a representative axoneme is highlighted in the panels *ii–iv*, with error bars indicating the standard error of the mean calculated by hexadecimation. (B) Equivalent plots to (A) for *mbo2* axonemes. (C) Beat shapes of one representative beat cycle of the wild type axoneme highlighted in panel A (left panel, data) and shapes reconstructed from the superposition of the static and fundamental modes, neglecting all higher harmonics. The progression of shapes through the beat cycle is represented by the rainbow color code (see inset). (D) Same as (C) for an *mbo2* mutant axoneme.

In our model, the response coefficients are small at zero frequency. This is because the observed, nearly constant static curvature implies that the static component of the force ($f_0$) is approximately zero all along the length except near the distal end of the axoneme (see *Equation 4*).

The motor force equation (*Equation 6*) together with the equation of motion (*Equation 5*), define a dynamical system, which can become unstable and produce spontaneous oscillations (*Jülicher and Prost, 1997*; *Camalet et al., 1999*). At the critical point, these oscillations are periodic and we only retain the fundamental mode ($n = 1$). These critical-point oscillations constitute the predicted beat waveforms of our two-dimensional model (see last paragraph of *Appendix 1*).

## Normal-force control

We conclude the theory section with a discussion of the normal force. Unlike the sliding control and curvature control mechanisms in which the motor force depends linearly on the tangent angle (and its derivatives), the feedback in the normal-force model is non-linear. This is because the normal force is the product of the integrated sliding force and the curvature, $f_\perp = F\dot{\psi}$ (*Appendix 1*,

Equation 16), both of which depend on arc-length. However, under the assumption that the static curvature is constant, $\dot{\psi}_0 = C_0$, the normal force can be linearized, simplifying the solution of the dynamical equations. Expanding $F$ and $\dot{\psi}$ into their static and fundamental Fourier modes, and using the static force balance $aF_0 = \kappa C_0$, we obtain the following expression for the fundamental mode of the normal force:

$$f_{\perp,1} = C_0\left(F_1 + \kappa\dot{\psi}_1/a\right), \tag{7}$$

where $F_1$ is the fundamental mode of the integrated sliding force (Appendix 1, Equation 19). Thus, the force is linear in the curvature and the integrated motor force. Equation 7 vanishes for symmetric beats in which the static curvature is zero ($C_0 = 0$). The important implication is that for symmetric beats, there is no reciprocal inhibition across the axoneme, unlike sliding and curvature control. This is related to the property that static bends produce normal forces that always tends to separate filaments, independent of the sign of the bend (Mukundan et al., 2014). Thus the static curvature, $C_0 \neq 0$, of the *Chlamydomonas* beat opens a way for regulation by normal forces, something impossible in symmetrically beating cilia, of which sperm is an approximate example (Riedel-Kruse et al., 2007).

## Results

### Quantification of the beat of wild type and *mbo2* cilia

To test the different mechanisms of beat regulation, we measured the flagellar beating waveforms in wild type and *mbo2* axonemes with high temporal and spatial precision (Materials and methods and Figure 2i-ii). We tracked trajectories of 20 points along the arc-length of the axoneme as a function of time over up to 200 beat cycles (Figure 2iii). Copies of the movies and the extracted tangent angles are available (see Sartori et al., 2016). The uncertainty of the position in $\mathrm{xy}$ space was $\approx 5$ nm and the uncertainty in the tangent angle was $\approx 20$ mrad (Figure 2iv-v). The latter corresponds to a sliding displacement between adjacent doublet microtubules of only 1.3 nm.

Because the beat of *Chlamydomonas* is periodic in time, it is convenient to decompose the tangent angle $\psi(s,t)$ into Fourier modes $\psi_n$ (Equation 2). Before doing so, we note that wild type *Chlamydomonas* axonemes swim counterclockwise in circles at a slow angular rotation speed $\omega_{rot} \approx 30 \mathrm{\ rad/s}$ (Figure 2Ai,iii). While the effect of this rotation is small for a single beat it becomes large for a long time series. Before performing the Fourier decomposition we therefore subtracted $\omega_{rot}t$ from the tangent angle $\psi(s,t)$ (for simplicity we use the same notation for $\psi(s,t)$ and $\psi(s,t) - \omega_{rot}t$). The power spectrum of the tangent angle (averaged over the flagellar length) shows clear peaks at harmonics of its fundamental frequency (Figure 3Ai). Because the peak at the fundamental frequency ($n = 1$) accounts for $90\%$ of the total power, we neglected the higher harmonics (n = 2,3,4, ...) for reconstructing the flagellar shape. We found that using just the $n = 0$ and $n = 1$ modes gave excellent reconstitutions of both the wild type and *mbo2* beats (Figure 3C–D). Thus, the static and fundamental modes provide a good description of the beats.

The amplitude of the static mode ($n = 0$) and the amplitude and phase of the fundamental mode ($n = 1$) are shown in (Figure 3ii–iv). The main difference between the wild type and mutant axonemes comes from the static mode, $\psi_0$ (Figure 3ii; Eshel and Brokaw, 1987; Geyer et al., 2016). For wild type axonemes, $\psi_0$ decreased approximately linearly over arc-length. This corresponds to an approximately constant static curvature $\approx 0.25$ rad/μm, and indicates that the time-averaged shape is close to a semi-circular arc of radius $\approx 4$ μm. The static curvature of wild type axonemes leads to the highly asymmetric waveform. In contrast, *mbo2* mutant axonemes have a small static mode, with a curvature $\approx 0.025$ rad/μm, corresponding to an approximately symmetric waveform.

In comparison to the large differences in the static mode between wild type and mutant axonemes, the fundamental modes, $\psi_1$, are similar. The amplitude of $\psi_1$ is roughly constant and has a characteristic dip in the middle (Figure 3iii). The argument of $\psi_1$, which determines the phase of the wave, decreases at a roughly constant rate in both cases (Figure 3iv), indicating that the beat is a traveling wave. Because the total phase shift is about $-2\pi$, the wavelength of the beat is approximately equal to the length of the axoneme. Thus, both wild type and mutant axonemes have

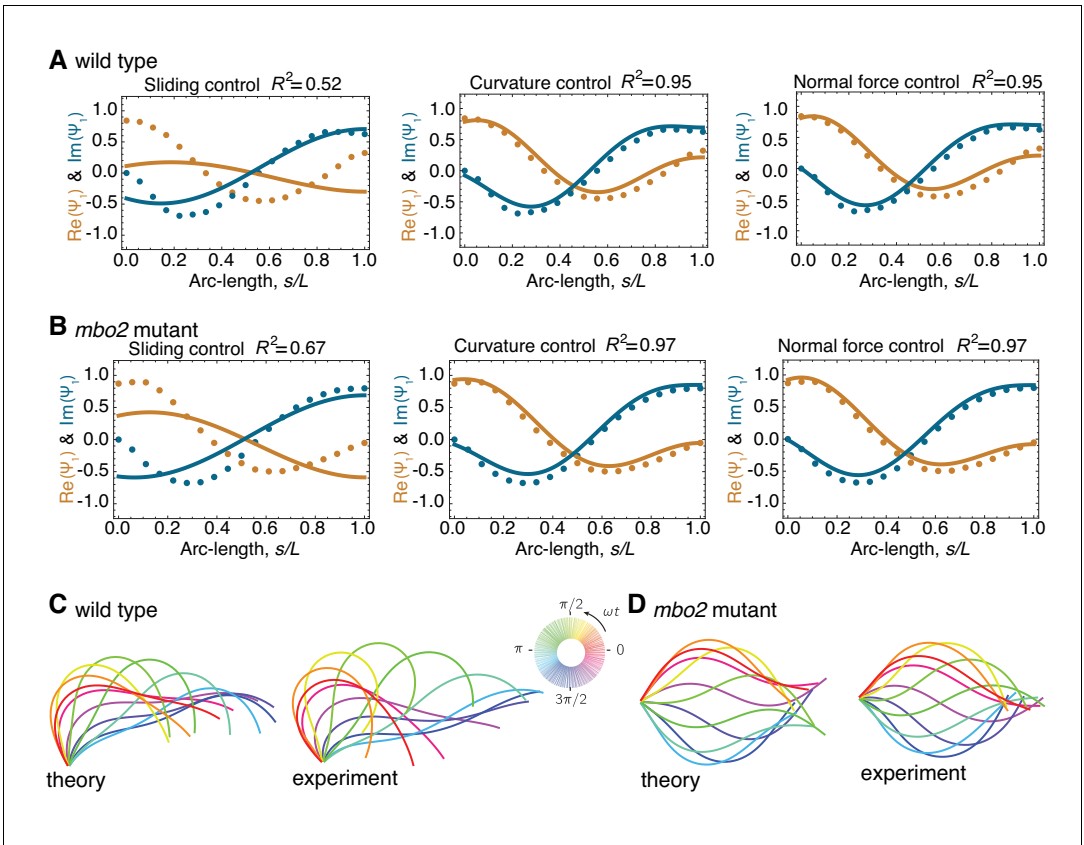

**Figure 4.** Comparison of theoretical and experimental beating patterns. (**A**) Comparison of the theoretical (lines) and experimental (dots) beating patterns of a typical wild type axoneme. The real and imaginary part of the first mode of the tangent angle $\psi_1(s)$ is plotted for beats resulting from sliding control, curvature control, and normal force control. (**B**) Analogous to (**A**) for *mbo2*. Note that here also curvature control and normal force control provide good agreement, but not sliding control. (**C**) and (**D**) Theoretical and experimental shape reconstruction in position space for the wild type and *mbo2* beats under curvature control.

approximately sinusoidal dynamic beats whose amplitudes dip in the middle of the axoneme and whose wavelengths are approximately equal to their lengths.

## Motor regulation in the axoneme: experiment versus theory

To gain insight into how molecular motors in the axoneme are controlled, we compared experimental beating patterns to those calculated from theory. In the *sliding-control* model the motor force depends only on sliding through the sliding coefficient $\chi = \chi' + i\chi''$. The single and double primes denote real and imaginary parts. $\chi'$ and $i\chi''$ describe the dependence of force on the sliding displacement and the sliding velocity. They are the elastic and damping components of the sliding response, respectively. Because the response must be active for oscillations to occur, we have $\chi', \chi'' \leq 0$ (**Machin, 1958**). In the sliding-control model, the motor force is independent of curvature and normal forces so the curvature coefficient ($\beta$) and the normal force coefficient ($\gamma$) were set to zero. In the *curvature-control* model, the curvature coefficient is non-zero ($\beta \neq 0$). In addition, we allow the possibility that the motors have a passive response to sliding (corresponding to elastic resistance to shear between adjacent doublets at the beat frequency), so that $\chi'>0$. The motors are not regulated by normal force ($\gamma = 0$). In the *normal-force* model, $\gamma \neq 0$. We again allow for the possibility that the motors have a passive response to sliding ($\chi'>0$). The motors are not regulated by curvature ($\beta = 0$)). Note that for backward traveling waves the signs of $\beta', \beta''$ and $\gamma', \gamma''$ change, but those of $\chi', \chi''$ do not.

We tested the three motor models by adjusting the appropriate response coefficients, together with the basal stiffness ($k_b$) and the basal damping coefficient ($\xi_b$), to obtain the closest fit of their predictions to the dynamic mode. The fitting procedure is described in *Appendix 3* using the mechanical parameters described in *Appendix 5*. The result of a typical fit for wild type axonemes is shown in *Figure 4A*. The real and imaginary parts of the fundamental mode $\psi_1(s)$ – corresponding to the cosine and sine components of the waveforms – agree well with the data in the cases of curvature control and normal-force control, but not for sliding control. In the latter case, the real and imaginary parts of the predicted mode are in anti-phase (*Figure 4A*, left panel), corresponding to a standing wave and contradicting the observed propagating wave. The **xy** representation of the beating pattern predicted by the curvature-control model agrees well with the experimental beating pattern reconstructed from the static and fundamental modes (*Figure 4C*). The good agreement reinforces the conclusion from *Figure 4A* that the curvature control model accords with the experimental data for wild type axonemes. Similar good agreement for wild type axonemes was found with the normal-force model. *Table 1* summarizes average parameters resulting from the fits to data from 9 wild type axonemes.

We compared the theory to the dynamic modes measured from the *mbo2* mutant, where the static curvature is reduced by at least one order of magnitude compared to wild type beats (*Figure 3*). The results were similar to those of wild type (*Figure 4B*): sliding control could not produce bend propagation, while curvature and normal force control were in good agreement with the experimental data. The parameters obtained from the fit of *mbo2* beats are given in *Table 2*. We also fit the model to the wild type waveforms in which the static curvature had been subtracted. A good fit to the curvature control model was obtained, but not to the sliding control model. Thus, the fundamental mode was well fit by the curvature and normal-force models but not sliding control.

## Regulation of the beat by sliding

The sliding-control model provides a poor fit to the observed beating patterns. This can be understood using three different, but related, arguments. First, in the plane-wave approximation, $\psi_1 = \exp(-2\pi i s/\lambda)$, the wavelength satisfies

$$i\omega\xi_n/\kappa = -(2\pi/\lambda)^4 - (2\pi/\lambda)^2 a^2 \chi/\kappa \quad . \tag{8}$$

Because the equation is unchanged when $\lambda \to -\lambda$, there are solutions for propagation from base to tip ($\lambda > 0$) and for propagation from tip to base ($\lambda < 0$). These two waves superimpose to form a standing wave, inconsistent with the observed traveling wave.

The second argument is that in the limit of very short axonemes ($L \to 0$), sliding control predicts that there will only be standing waves, irrespective of whether the boundary conditions are

**Table 1.** Parameters for beat generation in wild type axonemes.

|  |  | Sliding control | Curvature control | Normal-force control |
|---|---|---|---|---|
| Coefficient of determination | $R^2(\%)$ | $49 \pm 4$ | $95 \pm 1$ | $95 \pm 1$ |
| Sliding coefficient | $\chi', \chi''(\text{nN} \cdot \mu\text{m}^{-2})$ | $-12.2 \pm 3.0, -1.1 \pm 0.1$ | $19.8 \pm 3.3, 0$ | $13.2 \pm 3.6, 0$ |
| Curvature coefficient | $\beta', \beta''(\text{nN})$ | $0$ | $0, -6.5 \pm 0.4$ | $0$ |
| Normal-force coefficient | $\gamma', \gamma''$ | $0$ | $0$ | $0.12 \pm 0.05, 2.0 \pm 0.2$ |
| Basal impedance | $\chi_b', \chi_b''(\text{nN} \cdot \mu\text{m}^{-1})$ | $42.4 \pm 0.7, 2800 \pm 6900$ | $42.2 \pm 12, 2.2 \pm 0.7$ | $80 \pm 115, 13000 \pm 34000$ |
| Basal sliding (0th mode) | $\Delta_{b,0}$ (nm) | $-41 \pm 41$ | $-36 \pm 12$ | $-144 \pm 196$ |
| Basal sliding (1st mode) | $|\Delta_{b,1}|$ (nm) | $23 \pm 19$ | $22 \pm 5$ | $12 \pm 11$ |

The values reported are mean and standard deviation calculated from 9 axonemes. The average static curvature is $C_0 = -0.232 \pm 0.009 \; \mu\text{m}^{-1}$, the length $L = 11.7 \pm 0.4, \mu m$, and the frequency $\omega = 427 \pm 19$ rad/s (68 Hz). Note that, for curvature controlled beats, using the value of $\beta$ and an estimate curvature of $0.1 \, \text{rad}/\mu m$ results in a sliding force of $\approx 700 \, \text{pN}/\mu m$. Since the motor density of an active half of the axoneme is $\approx 500 \, \mu\text{m}^{-1}$ the individual motor force is $\sim 1 \, \text{pN}$. For the case of normal force control, the sliding force generated by the motors is of the order of the normal force that they experience, since $|\gamma| \approx 2$.

**Table 2.** Parameters for beat generation in *mbo2* mutant axonemes.

| | | Sliding control | Curvature control | Normal-force control |
|---|---|---|---|---|
| Coefficient of determination | $R^2$ (%) | $72 \pm 5$ | $95 \pm 1$ | $96 \pm 1$ |
| Sliding coefficient | $\chi', \chi''$ (nN·$\mu$m$^{-2}$) | $-15.0 \pm 1.0, -0.4 \pm 0.1$ | $17.4 \pm 2.5, 0$ | $10.5 \pm 4.3, 0$ |
| Curvature coefficient | $\beta', \beta''$ (pN) | 0 | $0, -0.66 \pm 0.4$ | 0 |
| Normal-force coefficient | $\gamma', \gamma''$ | 0 | 0 | $1.52 \pm 1.52, 32 \pm 25$ |
| Basal impedance | $\chi'_\mathrm{b}, \chi''_\mathrm{b}$ (nN·$\mu$m$^{-1}$) | $30.6 \pm 14.1, 1.6 \pm 0.7$ | $2.1 \pm 2, 2.3 \pm 1.7$ | $2.3 \pm 0.3, 13.6 \pm 6.0$ |
| Basal sliding (0th mode) | $\Delta_{\mathrm{b},0}$ (nm) | $-6 \pm 6$ | $-73 \pm 44$ | $-64 \pm 40$ |
| Basal sliding (1st mode) | $|\Delta_{\mathrm{b},1}|$ (nm) | $51 \pm 10$ | $60 \pm 11$ | $77 \pm 28$ |

Values are averages and standard deviations for 9 axonemes. The static curvature was $C_0 = -0.0276 \pm 0.005$ $\mu$m$^{-1}$, the length $L = 9.2 \pm 0.3$ $\mu m$, and the frequency $\omega = 176 \pm 44$ rad/s, (28 Hz). Note that the values of $\gamma$ in normal force control are very spread out and compared to the wild type fits. In fact, in one case we obtained $\gamma \approx 80$, indicating that motors must amplify the normal force they sense by almost two orders of magnitude. The values for curvature control are very similar to those of wild type fits..

symmetric or not (*Camalet and Jülicher, 2000*). *Chlamydomonas* axonemes are short in the sense that are much shorter than the critical length:

$$\ell = 2\pi \left( \frac{\kappa}{\xi_\mathrm{n} \omega} \right)^{1/4}, \tag{9}$$

which is $26 \, \mu$m using the *Chlamydomonas* parameters (*Appendix 5*). Based on this limit we only expect standing waves.

The third argument, is a generalization of the second. Though short, *Chlamydomonas* axonemes have a non-zero length. We therefore computed the bend propagation speed according to $v = \int_0^L |\psi_1|^2 \partial_s \arg \psi_1 \mathrm{d}s$. Using the amplitude and phase of the experimental data for *Chlamydomonas*, $|\psi_1| \sim 0.68$ (*Figure 2Aiii* and (*Geyer et al., 2016*) and $\arg \psi_1 = -2\pi s/L$ (*Figure 2Aiv*), the bend propagation speed is $\approx 3$. By contrast, the bend propagation speed predicted by the sliding control model is only $0.005 \ll 3$. Thus, the sliding-control model predicts a bend propagation speed much lower than observed.

That the low bend propagation speed is due to the short length of the *Chlamydomonas* axoneme can be appreciated by plotting the predicted speed (normalized by the measured speed of *Chlamydomons*) against length $L$ (normalized by the critical length). *Figure 5A* shows that at short lengths, such as for *Chlamydomonas*, the wave propagation speed is very slow, whereas for lengths above the critical length, such as for sperm, the propagation speed is high.

Thus, there are several arguments for why sliding control does not work for the short flagella of *Chlamydomonas*.

## Regulation of the beat by normal forces

The normal-force model provides a good fit to the beating patterns of wild type and *mbo2* axonemes. However, there are two related arguments against the normal-force model. First, despite the similarities in the dynamics of the beats of wild type and *mbo2* axonemes (*Figure 3iii–iv*), the normal force model requires very different values for the response coefficient, $\gamma$, for wild type and *mbo2* axonemes (*Tables 1* and *2*). Second, the normal-force model applied to *mbo2* requires large differences in $\gamma$ from axoneme to axoneme, despite the similarity in the dynamics among the axonemes (*Figure 3Biii–iv*). To understand why this is the case, we plotted $\gamma$ as a function of the inverse of the curvature. The two are strongly correlated: $|\gamma| \propto |C_0|^{-1}$(*Figure 6A*). This correlation follows from *Equation 7*, which predicts that the dynamic component of the normal force is linearly proportional to the static curvature $C_0$. In other words, the normal-force model requires there be static curvature; if the static curvature were exactly equal to zero then the model would break down. However, the static curvature in *mbo2* axonemes is so small, as few as 3 degrees over the length of the axoneme, that it is likely to be residual and of no significance. It is therefore puzzling why the key control parameter would depend so strongly on a residual property. By contrast, the curvature-control

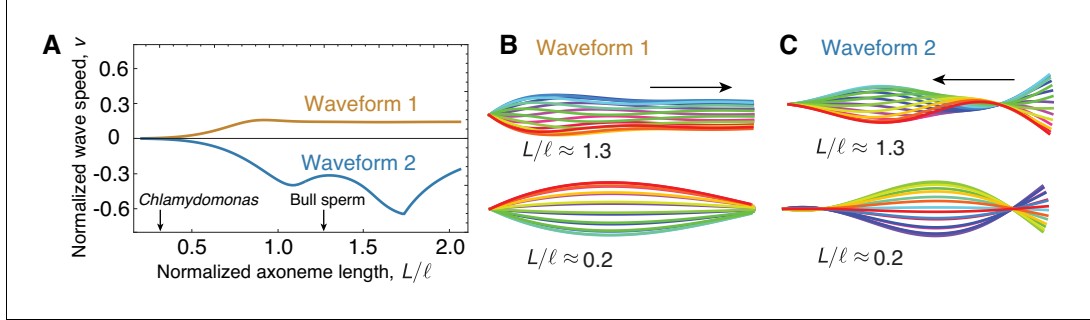

**Figure 5.** The role of length in sliding-regulated beats. (A) Wave speed versus relative length of the first two unstable waveforms of a freely swimming axoneme regulated by sliding. For short lengths, in the range of *Chlamydomonas*, the modes lose directionality and become standing waves. Long axonemes have directional waves that can travel either forward, as in waveform 1,or backwards, as in waveform 2. (B) Two examples of waveform 1 for long (top) and short (bottom) axonemes. Note that the short axonemes have standing waves. See (*Sartori, 2015*) for more details. (C) Beating patterns of waveform 2 for a long (top) and short (bottom) axonemes. In B and C arrows denote direction of wave propagation.

coefficient, $\beta$, is similar for wild type and *mbo2* axonemes and is independent of the static curvature (*Figure 6B*). Thus, we conclude that normal force is not a plausible parameter for controlling the ciliary beat.

## Regulation of the beat by curvature

The curvature control model provides a good fit to the experimental data for both wild type and *mbo2* axonemes (*Figure 4A and B*, middle panel). Initially, we fitted the data using non-zero values for both the sliding control parameters ($\chi'$, $\chi''$) and for both the curvature control parameters ($\beta'$, $\beta''$). We found that the best fit values of $\chi''$ and $\beta'$ were not significantly different from zero. Furthermore, the quality of the fits were as good when we set both to zero. We therefore took $\chi'' = \beta' = 0$ (*Tables 1* and *2*). We present an argument in the Discussion for why these two parameters are expected to be zero. Thus, the curvature control model is specified by just two free parameters, the sliding elasticity between doublet microtubules at the beat frequency ($\chi'$) and the rate of change of axonemal curvature ($\beta''$) (note that the parameters, $\chi'_b$ and $\chi''_b$, which characterize the stiffness and viscosity at the base respectively, are determined once $\chi'$ and $\beta''$ are specified in order to satisfy the boundary conditions [final paragraph of *Appendix 2*]).

The average values of $\chi'$ and $\beta''$ varied little between wild type and *mbo2* mutant axonemes (compare the third column of *Table 1* with that of *Table 2*, see also *Figure 6B*). This accords with the observation that there is little difference in the dynamical properties of the beat between wild type and *mbo2* axonemes. Furthermore, the standard deviations of $\chi'$ and $\beta''$ are small, indicating that there is little variation from axoneme to axoneme. Thus, the tight distribution of values of the parameters in the model reflects the similarity in the observed shapes in different axonemes. In other words, $\chi'$ and $\beta''$ are well constrained by the experimental data.

To understand what aspects of the experimental data specify these two parameters, we performed a sensitivity analysis on $\chi'$ and $\beta''$. In *Figure 7A* we show a density map of the mean square distance $R^2$ between the theoretical waveforms and a reference experimental beating pattern as a function of $\chi'$ and $\beta''$. The red ellipse delimits a region of good fit in which $R^2 > 0.90$. This region closely coincides with the region where $\chi'_b$ and $\chi''_b$ are both positive, which is delimited by the central pair of black lines. This is important because negative values of the basal parameters imply an active process at the base, which would result in a whip-like motion of the axoneme, as noted by Machin (*Machin, 1958*). From the shapes of beating spermatozoa, Machin argued against such an active base.

We systematically varied $\chi'$ and $\beta''$ parallel and perpendicular to the long axis of the ellipse. Moving perpendicular into the region of active base indeed results in whip-like beats, with a larger amplitude at the base (*Figure 7B*, blue circles). This argues against a basal, whip-like driving of the

**Figure 6.** Scaling of response coefficients with static curvature. (A) Because the normal force $f_{\perp,1}$ is proportional to the static curvature $C_0$, the normal force response coefficient $\gamma$ is inversely proportional to the curvature. Red symbols are *mbo2* and green symbols are wild type. (B) The curvature control response coefficient $\beta''$ is independent of the static curvature, and remains constant even for a fifty-fold change in static curvature.

motion of *Chlamydomonas* axonemes. Moving parallel affects the amplitude of the beat, with the middle-dip becoming more or less prominent (*Figure 7B*, green circles). Thus, the arc-length dependence of the amplitude of the fundamental mode constrains the value of the response coefficients and the sign of the basal response.

To better understand the cell-to-cell variability we plotted data from all the axonemes in the $(\chi', \beta'')$ space (*Figure 8A*). Points scatter mainly along the long axis of the ellipse, where there is a large region of small shape variation. We consistently saw a shift perpendicular to the long axis between the wild type and *mbo2* axonemes. This variation correlates with the difference in normalized lengths between wild type and mutant axonemes (*Figure 8B*), suggesting a possible dependence of the response coefficients on length and/or frequency.

A striking difference between the wild type and *mbo2* axonemes is that the curve-fitting indicates that the basal stiffness in the mutant is about 20-fold smaller than in the wild type (*Table 1* and *2*). This softening at the base is associated with a larger basal sliding in the mutant. Whether this difference causes the difference in beat frequency (the mutant beats more slowly) or whether it is a consequence of the shorter lengths of the mutant axonemes (*Figure 8C*) will require additional study.

## Discussion

In this work we imaged isolated axonemes of *Chlamydomonas* with high spatial and temporal resolution. We decomposed the beating patterns into Fourier modes and compared the fundamental mode, which is the dominant dynamic mode, with theoretical predictions of three motor control mechanisms built into a two-dimensional model of the axoneme (*Figure 1*). The sliding control model provided a poor fit to the experimental data. We argued that the reason for this is that sliding control cannot produce wave propagation for axonemes as short as those of *Chlamydomonas* (*Figure 5*). While the normal-force model (also termed the geometric clutch model) provided good fits to the experimental data, it relies on the presence of static curvature (*Bayly and Wilson, 2015*), which varies greatly between the *mbo2* and wild type axonemes. As a result of this large difference in static curvature, the control parameters in this model had to be varied over a wide range to fit the data from the different axonemes (*Figure 6*). Because the waveforms of *mbo2* and wild type axonemes have similar dynamic characteristics, such variation in the control parameter seems implausible and we therefore argue against regulation by normal forces in the two-dimensional model. Finally, the curvature-control model provided a good fit to the experimental data with similar parameters for *mbo2* and wild type axonemes. Thus, we conclude that only the curvature-control model is fully consistent with our experimental data.

A potential caveat of these conclusions is that the model used here is two-dimensional. Importantly, in order to simplify the geometry, the model only contains one pair of filaments. While this captures the essential features of the sliding control and curvature control models, it oversimplifies the normal-force model, because in the three-dimensional axoneme there are radial and transverse forces acting on the doublets as the axoneme bends. Yet the two-dimensional model does not

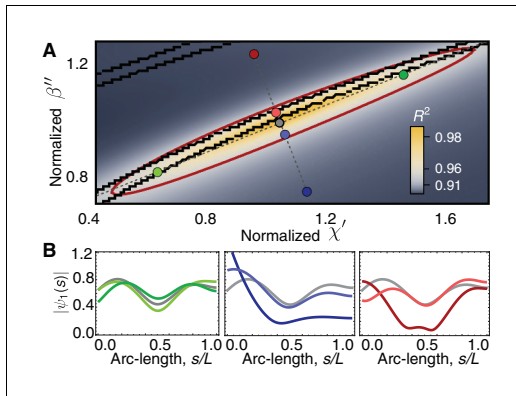

**Figure 7.** Phase space of curvature control. (**A**) Heat map of the mean-square distance $R^2$ between the theoretical and a reference experimental beat as a function of the sliding response coefficient $\chi'$ and the curvature response coefficient $\beta''$. The ellipsoid delimits the region with $R^2 = 0.90$. Each pair of nearby black lines delimits a region with a passive base. Moving along the long axis (green circles) affects the amplitude dip in the midpoint of the axonemem, left panel in (**B**). Moving along the short axis towards the region of active base results in waveforms with a large amplitude at the base (blue and red circles), central and right panels in B. The axis in A is normalized by the reference fit, such that $(\chi' = 1, \beta'' = 1)$ corresponds to the highest value of $R^2$.

distinguish between them. To bridge this gap, in other work (*Sartori et al., 2015*) we developed a full three-dimensional model of the axoneme to calculate the radial and transverse stresses. The three-dimensional model shows that even when there is a static curvature (without twist), normal (transverse) forces are not antagonistic across the centerline and therefore cannot serve as a control parameter for motors.

## Relation with past work

Earlier results showed that sliding control can account for the beating patterns of sperm (*Camalet and Jülicher, 2000*; *Riedel-Kruse et al., 2007*; *Brokaw, 1975*). This is consistent with the present results because the bull sperm axoneme is approximately five times longer than the *Chlamydomonas* axoneme and we have shown that sliding control can lead to bend propagation in long axonemes (*Figure 5* and see *Brokaw, 2005*). Thus, it is possible that different control mechanisms operate in different cilia and flagella, with sliding control being used in longer axonemes and curvature control being used in shorter ones. However we do note that curvature control models can account for the bull sperm data (*Riedel-Kruse et al., 2007*), as well as data from other sperm (*Brokaw, 2002*; *Brokaw, 1985*; *Bayly and Wilson, 2015*), so there is no strong morphological evidence favoring either sliding or curvature control in sperm. The normal-force model produces beating patterns that resemble those of sperm (*Lindemann, 1994b*; *Bayly and Wilson, 2015*; *Bayly and Wilson, 2014*). However, these models rely on there being an asymmetry which is small and variable in sperm, arguing against normal-force control. Thus, the curvature control model, unlike the other two models, robustly describes symmetric and asymmetric beats in short and long axonemes, and could serve as a 'universal' regulator of flagellar mechanics.

## Dynamic curvature control as a mechanism for motor regulation

An unexpected feature of our curvature control model is that the motor force depends only on the time derivative of the curvature. This follows from the fact that the curvature response function $\beta$ has no real part (*Table 1* and *2*, see Theoretical Model section). Such a model is fundamentally different from the current views of curvature control, in which motors are thought to respond to instantaneous curvature (*Brokaw, 1972*; *Brokaw, 2002*; *Brokaw, 2009*) and not to its time derivative.

While motors can respond to time derivatives of sliding displacement through their force-velocity relation, it is hard to understand how a similar mechanism could apply to curvature. One possibility is that there is a curvature adaptation system analogous to that of sensory systems, like the signaling pathway of bacterial chemotaxis (*Macnab and Koshland, 1972*; *Yi et al., 2000*; *Shimizu et al., 2010*). In an adaptation mechanism, curvature (or motor activity) would be 'remembered', and the average curvature (or motor activity) over past times would in turn down-regulate the activity of the motors on a long time-scale. Such regulation could occur, for example, via phosphorylation sites in the dynein regulatory complex or the radial spokes (*Witman, 2009*; *Smith and Yang, 2004*; *Porter and Sale, 2000*). Just as methylation of the chemoreceptors of bacteria modifies their ligand affinity, phosphorylation of regulatory elements within the axoneme could modify the motor sensitivity to curvature over times long compared to the period of the beat.

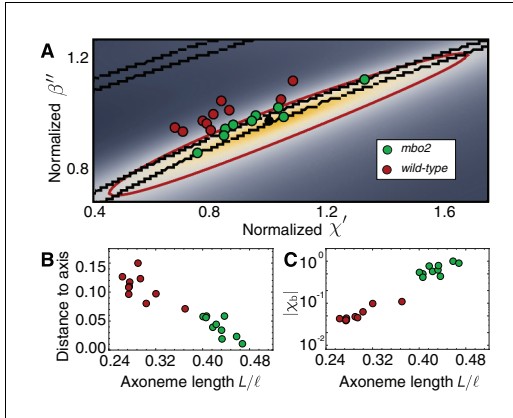

**Figure 8.** Axonemal variability in phase space. (**A**) Circles represent values obtained from fits for each of the axonemes, green corresponds to wild type and red to *mbo2*. In the background we have the same heat map as in *Figure 7*. Note that *mbo2* points lie away from wild type circles in the direction of the short axis of the ellipse. All values are normalized by those of the reference fit used also to normalize the heat map axis. (**B**) The distance of the circles to the long axis of all fits shows a clear correlation with normalized axonemal length. Note also that *mbo2* axonemes are systematically shorter than wild type axonemes. (**C**) The basal compliance ($\chi_b$) also correlates with the normalized length, resulting in a high value for wild type axonemes, which are longer. The values are normalized by the value for a reference axoneme.

The molecular mechanisms underlying curvature sensing are unknown. A difficulty with dyneins directly sensing microtubule curvature is that the strain in a curved microtubule (radius of curvature 4 µm) is very small ($\approx 1\%$), corresponding to a sub-angstrom strain in a tubulin dimer. Such a small strain would be difficult for an individual dynein microtubule-binding domain to detect, though this difficulty could be circumvented if dynein binding were cooperative, as found for microtubule curvature-sensing by doublecortin (*Bechstedt et al., 2014*). On the other hand, indirect curvature-sensing mechanisms that rely on the central pair pathway (*Witman, 2009*) are difficult to reconcile with mutants missing the central pair and radial spokes (*Yagi and Kamiya, 2000*; *Frey et al., 1997*). Thus, our findings highlight the question of how curvature might be sensed in the axoneme.

## Independence of static and dynamic waveform components

Our dynamic curvature control model adds to the view that dynamic and static components of the beat are regulated independently. The problem with models in which dynein activity is regulated by the instantaneous value of the curvature is that both the static and dynamic components of the beat would contribute to regulation and hence the dynamic component of the waveform would be highly dependent on the static component (*Sartori, 2015*). Yet the dynamic beats of wild type and *mbo2* are similar, as also noted in (*Eshel and Brokaw, 1987*). Our dynamic curvature control model provides a solution to this problem because static curvature is 'adapted' away. We now bring together several lines of evidence supporting the notion that the static and dynamic modes are separable in their origin and in their affect on the beat.

1. Dynamic and static components of the beat can exist independently of each other. This is evidenced by the existence of bent, non-motile cilia at low ATP concentrations on the one hand, as well as symmetrically beating mutants on the other (*Geyer et al., 2016*).
2. The waveform of *mbo2* has a fundamental dynamic mode similar to that of wild type, *Figure 3 Aiii and Biii*. However, the static mode is absent in the former, *Figure 3 Aii and Bii*. The same also holds for the two beating modes of the uniflagellar mutant (*Eshel and Brokaw, 1987*). Thus, altering the static mode of *Chlamydomonas* has little effect on the dynamic mode.
3. The dynamic motor response coefficients are largely independent of the asymmetry, and very similar for *mbo2* and wild type axonemes, (*Figure 6 A*).

If the dynamic and static modes are indeed independently controlled, the dynamic motor response is robust to changes in the asymmetry. This has important biological implications: power generation (the beat) and steering (the asymmetry) can be independently controlled so that the swimming direction can be adjusted without having to alter the motor properties. We note, however, that the molecular origin of asymmetry is not known nor is the mechanism by which the *mbo2* mutation leads to symmetric beats (see *Geyer et al., 2016* for a discussion of possible mechanisms).

## Phase of motor activity during the beat

Our work predicts how the timing of motor activity drives the bending of the axoneme during the beat. The simplest way to understand the spatio-temporal relationship between motor force, $f$, and axoneme curvature, $\dot{\psi}$, is to use the plane-wave approximation $\psi(s, t) = sin(\omega t - 2\pi s/\lambda)$, where $\omega$ is the frequency in radians per second and $\lambda$ is the wavelength in microns. Differentiation with respect to time $t$, leads to a phase advance by $\pi/2$ (90 degrees or a quarter of a cycle). This can be appreciated by noting that the cosine function (the derivative of sine) reaches its maximum before the sine function (in the complex representation, $exp[i(\omega t - 2\pi s/\lambda)]$, differentiation with respect to time leads to multiplication by $i$, which is equivalent to a phase lead of $\pi/2$, since $i = exp(i\pi/2)$). For a traveling wave moving from base to tip (i.e. moving in the direction of increasing arc-length, $s$), the wavelength is positive and differentiation with respect to arc-length leads to a phase delay by $\pi/2$. This can be appreciated by noting that $-cos(s)$, the derivative of $sin(-s)$, reaches its maximum after the original function. Thus, time derivatives lead and spatial derivatives lag.

Using these differentiation rules, we can represent the phase relations between the axonemal parameters on a phase plot (*Figure 9A*). For the reference phase, we use the sliding displacement, $\Delta$, or the tangent angle $\psi$, which are proportional to each other in the plane-wave approximation ($\Delta = a\psi$): we define them (arbitrarily) to zero degrees (compass bearing east, E). The sliding velocity $\partial_t \Delta$ has a phase lead of $\pi/2$ (i.e. compass bearing N on the phase plot). The curvature, the derivative of tangent angle with respect to arc-length, $\partial_s \psi = \dot{\psi}$, has a phase delay of $\pi/2$ (i.e. S on the phase plot). The key parameter is motor force, which, in the curvature-control model, is $f_1 = \chi' \Delta_1 + i\beta'' \partial_s \psi_1 \approx [a\chi' + \beta''(2\pi/\lambda)]\psi_1$. Using the parameters for wild type axonemes in *Table 1*, the sliding stiffness $\chi' = +20\,nN/\mu m^2$ and the dynamic curvature coefficient $\beta'' = -6.5\,nN$, together with $a = 0.066\,\mu m$ and $\lambda = L = 11.7\,\mu m$, we find that the absolute magnitude of the first term in the square bracket is smaller than the second term, which is negative. Thus, in the plane-wave approximation, the sign of the force is opposite that of the tangent angle and so the motor force is out of phase with the tangent angle and sliding (compass bearing W).

An exact calculation shows that the phases predicted by the curvature control model are similar to those of the plane-wave approximation. Thus, the tangent angle (*Figure 9A*, gray arrow) lags slightly behind the sliding displacement (*Figure 9A*, black arrow, E) due to the delay associated with the basal compliances. Furthermore, the curvature (*Figure 9A*, blue arrow) lags the plane-wave curvature (S) due to the basal compliance and because the wavelength has a small imaginary component. Likewise, the motor force (*Figure 9A*, red arrow) lags slightly behind the plane-wave force (W). Thus, the exact theory shows that curvature leads the motor force by approximately one quarter of a cycle and is nearly out of phase with the sliding displacement. The time series for the various parameters are plotted in *Figure 9B*.

From these phase relations, we can use the curvature-control model to predict the activity of the motors in relation to the curvature of the axoneme. These predictions can then be compared to structural studies, such as cryo-electon microscopy. Because the motor force lags the curvature (*Figure 9B*), the motor force is in phase with the spatial derivative of the curvature. Thus, the motor force is positive in the region marked green in *Figure 9D* and negative in the blue region. In other words, the motors whose bases are statically attached to the upper filament are actively interacting with the lower filament with their microtubule-binding domains to drive the bend that will develop at this place as the wave travels towards the distal tip. The motors with opposite polarity (i.e. on the opposite side of the axoneme) will be active proximal to the bend. Such a relative phase of the motor activity with respect to curvature is a consequence of the dynamic curvature mechanism: the force that generates the curvature (*Figure 9C*, green) is activated by the rate of change in curvature with a sign ($\beta'' < 0$) consistent with our sign convention. Note that if these predictions are to be compared to experiments, then the static curvature needs to be subtracted.

## Efficiency and energetics

Our finding that the motor force is nearly out of phase with the sliding displacement (and tangent angle) shows that the *Chlamydomonas* flagellum does not operate close to optimal efficiency. The maximally efficient phase was defined as the phase of the motors that maximized the swimming speed while minimizing the elastic and viscous dissipation (i.e. minimizing energy consumption) (*Machin, 1958*). Machin showed that for a plane wave, the optimum occurs when the phase of the

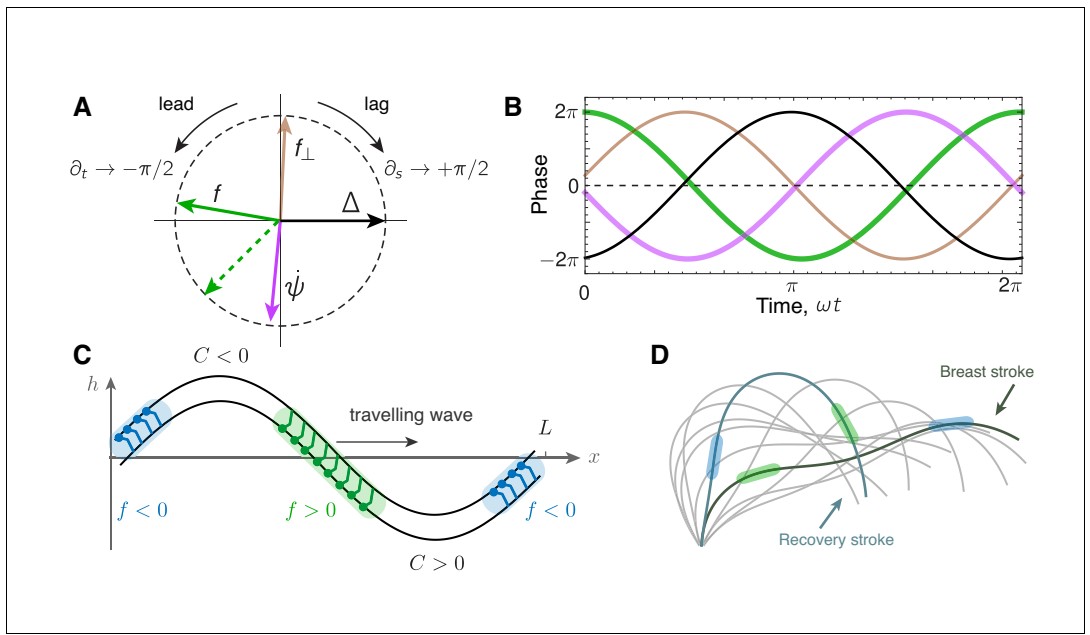

**Figure 9.** Phase delays and active force distribution on the axoneme. (A) Polar representation of the phases of sliding $\Delta$, curvature $\dot{\psi}$, normal force $f_\perp$ and the sliding force $f$ for a beating axoneme regulated by curvature. For a plane wave, $\psi(s,t) = \sin(\omega t - 2\pi s/\lambda)$, a time derivative rotates the phase counter-clockwise through $\pi/2$ radians, while a spatial derivative rotates the phase clockwise by $\pi/2$ radians. Deviations from the compass points (E, S, W, N) are due to deviations of the axonemal beat from a plane wave. Machin's prediction for an optimum flagellum is shown as the red dashed line. (B) Time evolution of the quantities in A.(C) Illustration of the local sliding force for a shape of a symmetrically beating axoneme (e.g. *mbo2*). The regions of motor activity on the upper and lower filament are highlighted in blue and green respectively. (D) Asymmetrically beating wild type axoneme with two shapes highlighted during the breast and recovery strokes. The arrows represent direction of motion of the axoneme, and the colored patches represent the local sliding force on the respective filaments (see panel C and *Figure 1*).

motors is at 225 degrees (compass bearing SW), considerably different from our predicted motor phase, which is <180 degrees. Thus, unlike sperm, which Machin calculated to be close to the optimum, the short cilia of *Chlamydomonas* deviate considerably from optimum efficiency.

The reason that *Chlamydomonas* deviates from optimum efficiency is that the elastic dissipation dominates over the viscous dissipation. Elastic dissipation arises from the straightening of the bent axoneme and the loss of the associated bending energy (for an asymmetric beat like that of *Chlamydomonas*, in which the static and dynamic modes have approximately equal amplitudes, the elastic loss is almost twice as great as that for a symmetric beat). Indeed, in the plane wave approximation, the ratio of the viscous to elastic dissipation is $(\pi/2)(L/\ell)^4 \ll 1$. This inequality, which was noted recently (*Chen et al., 2015*), holds for *Chlamydomonas* because *Chlamydomonas* axonemes are shorter than the critical length ($\ell$). By contrast, the inequality is $\gg 1$ for the much longer mammalian sperm axonemes (and $\approx 1$ for sea-urchin sperm). Note that the dependence of the critical length (*Equation 9*) on the mechanical parameters - drag coefficient and bending rigidity - is very weak due to the 1/4 power dependence. As a result, even if the bending rigidity were five times smaller than our estimate (*Pelle et al., 2009*), our argument would still be valid. Thus, the short length of *Chlamydomonas* axonemes has important implications for the energetics of the flagellar beat.

The reasoning behind the relatively small viscous dissipation also allows us to understand why the curvature mechanism is sensitive to the rate of change of curvature rather than the instantaneous value. *Equation 5* for a symmetric plane wave ($C_0 = 0$) regulated by curvature (using $f_1 = \chi a\psi + \beta\dot{\psi} = \chi a\psi + i\beta\psi 2\pi/\lambda$) gives

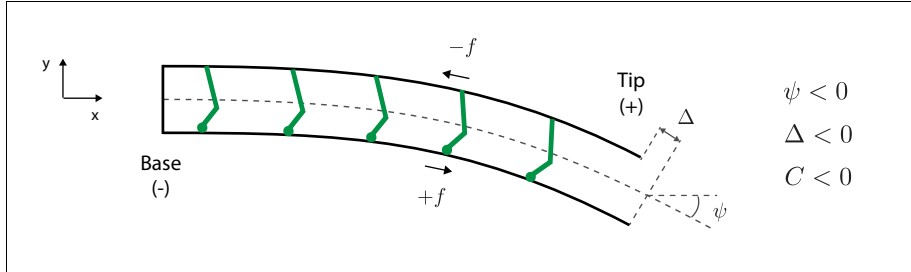

**Figure 10.** Sign convention for motors that step towards the base. See text for details

$$\xi_n \omega + a^2 \chi''(2\pi/\lambda)^2 - a\beta'(2\pi/\lambda)^3 = 0$$
$$\kappa(2\pi/\lambda)^4 + a^2 \chi'(2\pi/\lambda)^2 + a\beta''(2\pi/\lambda)^3 = 0 \tag{10}$$

The upper line of *Equation 10* shows that $\beta'$ produces an active motor force to counter the viscous dissipation due to the fluid and the inter-filament sliding. If the fluid damping coefficient is small, then we can have solutions with $\chi'' = \beta' \approx 0$. Furthermore, if the sliding response coefficient is small ($\chi', \chi''$ small) then $|\beta'/\beta''| \approx (\lambda/l)^4 \ll 1$, showing that the dynamic curvature response dominates over the instantaneous response. Thus, the dynamic curvature dependence is a consequence of the viscous forces being small compared to the elastic forces.

## Summary and outlook

We finish the discussion by noting that the short length of *Chlamydomonas* flagella, relative to the critical length $\ell = 2\pi(\kappa/\xi_n\omega)^{1/4}$ is key to understanding not just the energetics, but also the mechanism of motor control. Indeed, it was the shortness of the axonemes that implied that the sliding control model only generated a standing wave and so could not recapitulate the observed traveling wave. Furthermore, it was the shortness that gave rise to the imaginary curvature control coefficient, leading to the dynamic curvature-control mechanism. In addition, the shortness implies that the viscous dissipation is small compared to the elastic dissipation. These conclusions are robust because the power of 1/4 in the expression for the critical length makes it quite insensitive to the parameter values. The long length of sperm leads to quantitatively different properties (e.g the viscous and elastic energies are of similar magnitude, leading to efficient motion, as argued by Machin). Whether the long axonemes use the same dynamic curvature-control mechanism as short axonemes will require further study.

## Materials and methods

### Preparation and reactivation of axonemes

Axonemes from *Chlamydomonas reinhardtii* wild type cells (CC-125 wild type mt+ 137c, R.P. Levine via N.W. Gillham, 1968) and mutant cells that move backwards only (CC-2377 *mbo2* mt-, David Luck, Rockefeller University, May 1989) were purified and reactivated. The procedures described in the following are detailed in *Alper et al. (2013)*.

Chemicals were purchased from Sigma Aldrich, MO if not stated otherwise. In brief, cells were grown in TAP+P medium under conditions of illumination (2x75 W, fluorescent bulb) and air bubbling at 24°C over the course of 2 days, to a final density of $10^6$ cells/ml. Flagella were isolated using dibucaine, then purified on a 25% sucrose cushion and demembranated in HMDEK (30 mM HEPES-KOH, 5 mM MgSO$_4$, 1 mM DTT, 1 mM EGTA, 50 mM potassium acetate, pH 7.4) augmented with 1% (v/v) Igpal and 0.2 mM Pefabloc SC. The membrane-free axonemes were resuspended in HMDEK plus 1% (w/v) polyethylene glycol (molecular weight 20 kDa), 30% sucrose, 0.2 mM Pefabloc and stored at −80°C. Prior to reactivation, axonemes were thawed at room temperature, then kept on ice. Thawed axonemes were used for up to 2 hr.

Reactivation was performed in flow chambers of depth 100 µm, built from easy-cleaned glass and double-sided sticky tape. Thawed axonemes were diluted in HMDEKP reactivation buffer containing 1 mM ATP and an ATP-regeneration system (5 units/ml creatine kinase, 6 mM creatine phosphate) used to maintain the ATP concentration. The axoneme dilution was infused into a glass chamber, which was blocked using casein solution (from bovine milk, 2 mg/mL) for 10 min and then sealed with vacuum grease. Prior to imaging, the sample was equilibrated on the microscope for 5 min and data was collected for a maximum time of 20 min.

## Imaging of axonemes

The reactivated axonemes were imaged by either phase constrast microscopy (wild type axonemes) or darkfield microscopy (*mbo2* axonemes). Phase contrast microscopy was set up on an inverted Zeiss Axiovert S100-TV microscope using a Zeiss 63× Plan-Apochromat NA 1.4 Phase3 oil lens in combination with a 1.6× tube lens and a Zeiss oil condenser (NA 1.4). Data were acquired using a EoSens 3CL CMOS highspeed camera. The effective pixel size was 139 nm/pixel. Darkfield microscopy was set up on an inverted Zeiss Axiovert 200 microscope using a Zeiss 63× Plan achromat NA iris 0.7–1.4 oil lens in combination with an 1.25× tube lens and a Zeiss oil darkfield condenser (NA 1.4). Data were acquired using a pco dmaxS highspeed camera. In both cases, the illumination was performed using a Sola light engine with a 455 LP filter. Movies of up to 3000 frames were recorded at a frame rate of 1000 fps. The sample temperature was kept constant at 24°C using an objective heater (Chromaphor).

## High precision tracking of isolated axonemes

To track the axoneme in each movie frame with nm precision, the Matlab-based software tool FIESTA was used (*Ruhnow et al., 2011*). Prior to tracking, movies were background subtracted to remove static inhomogeneities arising from uneven illumination and dirt particles. The background image contained the mean intensity in each pixel calculated over the entire movie. This procedure increased the signal-to-noise ratio by a factor of 3 (*Alper et al., 2013*). Phase-contrast images were inverted; darkfield images were tracked directly.

The tracking algorithm FIESTA uses manual thresholding to determine the filament skeleton, which is then divided into square segments. During tracking, the filament position in each segment is determined independently. For tracking, a segment size of 733 nm (approximately 5x5 pixels) was used, corresponding to the following program settings: a full width at half maximum of 750 nm, and a "reduced box size for tracking especially curved filaments" of 30%. Along the arc-length of each filament, 20 equally spaced segments were fitted using two-dimensional Gaussian functions. Two examples of spline-fitted shapes are presented in *Figure 2Ai and Bi* superimposed on the image. The mean localization uncertainty of the center position of each of these segments was about 5 nm (*Figure 2Aiv and Biv*). For localization of the ends, the program uses a different fitting function, resulting in an increased uncertainty.

## Additional information

### Competing interests

FJ: Reviewing editor, *eLife*. The other authors declare that no competing interests exist.

### Funding

| Funder | Author |
| --- | --- |
| Yale University | Jonathon Howard |
| Max-Planck-Gesellschaft | Frank Jülicher |

The funders had no role in study design, data collection and interpretation, or the decision to submit the work for publication.

## Author contributions

PS, FJ, JH, Participated in conception and design, Analysis and interpretation of data, Drafted and revised the article; VFG, Participated in conception and design, Analysis and interpretation of data, Drafted and revised the article, Acquired data; AS, Participated in analysis and interpretation of data

## Author ORCIDs

Frank Jülicher, http://orcid.org/0000-0003-4731-9185
Jonathon Howard, http://orcid.org/0000-0003-0086-1196

# Additional files

## Major datasets

The following dataset was generated:

| Author(s) | Year | Dataset title | Dataset URL | Database, license, and accessibility information |
|---|---|---|---|---|
| Sartori P, Geyer VF, Scholich A, Jülicher F, Howard J | 2016 | Data from: Dynamic curvature regulation accounts for the symmetric and asymmetric beats of *Chlamydomonas* flagella | http://dx.doi.org/10.5061/dryad.0529j | Available at Dryad Digital Repository under a CC0 Public Domain Dedication |

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

## Appendices

# 1. Non-linear dynamics of the axoneme

The equations that describe the dynamics of the axoneme are obtained by balancing mechanical and fluid forces. We used a variational approach similar to that in *Camalet and Jülicher (2000)*, *Riedel-Kruse et al. (2007)*, *Sartori (2015)*. The work done on the filament pair when it is bent and tensed by the motor force $f$ is

$$U = \int_0^L \left[\frac{\kappa}{2}\dot{\psi}^2 + f\Delta + f_\perp(a'-a) + \frac{\Lambda}{2}(\dot{\mathbf{r}}^2-1)\right]\mathrm{d}s + \frac{k_\mathrm{b}}{2}\Delta_\mathrm{b}^2 \quad, \tag{11}$$

where $\kappa$ is the bending rigidity and $k_\mathrm{b}$ the stiffness of cross-linkers at the base. The normal force $f_\perp$ is a Lagrange multiplier that ensures that the interfilament spacing $a'$ is a constant. Similarly, $\Lambda$ is a multiplier that ensures the incompressibility constraint $\dot{\mathbf{r}}^2 = 1$, and is related to the tension in the centerline through $\tau = \Lambda + \kappa\dot{\psi}^2 - aF\dot{\psi}$, where $F$ is the integrated motor force (*Equation 3*).

The instantaneous mechanical force that the axoneme exerts on the fluid is given by $\delta U/\delta\mathbf{r}$, and calculating it requires computing $\delta\dot{\psi}$. From the relation $\mathbf{r}(s) = \mathbf{r}_0 + \int_0^s(\cos(\psi(s')), \sin(\psi(s')))\mathrm{d}s'$, where $\mathbf{r}_0$ is the position of the base, it follows that $\dot{\psi} = \mathbf{n}\cdot\dot{\mathbf{t}}$ and $\delta\dot{\psi} = \mathbf{n}\cdot\delta\ddot{\mathbf{r}}$. Using this, we arrive at $\delta U/\delta\mathbf{r} = \partial_s\left[(\kappa\ddot{\psi} - af)\mathbf{n} - \tau\mathbf{t}\right]$ (*Camalet and Jülicher, 2000*; *Sartori, 2015*). The net sliding force exerted at the base is $\delta U/\delta\Delta_\mathrm{b} = -F(0) + k_\mathrm{b}\Delta_\mathrm{b}$(*Mukundan et al., 2014*). To obtain the dynamics of the axoneme we balance these mechanical forces by the fluid friction $\hat{\Pi}\cdot\partial_t\mathbf{r}$ and the basal friction $\xi_\mathrm{b}\partial_t\Delta_\mathrm{b}$, which results in

$$\partial_t\mathbf{r} = -(\xi_\mathrm{n}^{-1}\mathbf{nn} + \xi_\mathrm{t}^{-1}\mathbf{tt})\cdot\partial_s\left[(\kappa\ddot{\psi} - af)\mathbf{n} - \tau\mathbf{t}\right] \tag{12}$$

$$\partial_t\Delta_\mathrm{b} = -\xi_\mathrm{b}^{-1}(k_\mathrm{b}\Delta_\mathrm{b} - F(0)) \quad, \tag{13}$$

We can calculate a dynamic equation for the tangent angle using that $\partial_t\dot{\mathbf{r}} = \mathbf{n}\partial_t\psi$, which results in

$$\partial_t\psi = \xi_\mathrm{n}^{-1}(-\kappa\ddddot{\psi} + a\ddot{f} + \dot{\psi}\dot{\tau} + \tau\ddot{\psi}) + \xi_\mathrm{t}^{-1}\dot{\psi}(\kappa\dot{\psi}\ddot{\psi} - af\dot{\psi} + \dot{\tau}) \quad. \tag{14}$$

This equation contains no information about the trajectory of the basal point $\mathbf{r}_0(t)$, which can be determined from the condition that the total force on the cilium vanishes (*Friedrich et al., 2010*; *Sartori, 2015*; *Johnson and Brokaw, 1979*).

The tension $\tau$ and normal force $f_\perp$ are obtained by imposing the corresponding constraints. For the case of the tension we take the time derivative of $\dot{\mathbf{r}}^2 = 1$. This gives $\mathbf{t}\cdot\partial_t\dot{\mathbf{r}} = 0$, where we can replace the dynamic equation for $\mathbf{r}$. For the normal force $f_\perp$ we use the force balance $\delta G/\delta a = 0$ (*Camalet and Jülicher, 2000*; *Mukundan et al., 2014*). The resulting equations are

$$\frac{\xi_\mathrm{n}}{\xi_\mathrm{t}}\ddot{\tau} - \dot{\psi}^2\tau = -\dot{\psi}(\kappa\ddot{\psi} - a\dot{f}) + \frac{\xi_\mathrm{n}}{\xi_\mathrm{t}}\partial_s[\dot{\psi}(af - \kappa\ddot{\psi})] \tag{15}$$

$$f_\perp = F\dot{\psi} \quad. \tag{16}$$

Because *Equation 16* plays a key role in normal-force control, we expand out the modes:

$$f_\perp = (F_0 + F_1 e^{i\omega t} + \dots)(\dot\psi_0 + \dot\psi_1 e^{i\omega t} + \dots) \tag{17}$$

$$= (\frac{\kappa C_0}{a} + F_1 e^{i\omega t} + \dots)(C_0 + \dot\psi_1 e^{i\omega t} + \dots) \tag{18}$$

$$= \frac{\kappa}{a} C_0^2 + C_0[F_1 + \frac{\kappa}{a}\dot\psi_1]e^{i\omega t} + F_1 \dot\psi_1 e^{2i\omega t}\dots \tag{19}$$

where the second term corresponds to *Equation 7*. To solve the dynamic equations we need to use boundary conditions. These represent force and torque balances at the ends of the filament pair. For the case of a freely swimming axoneme (no external forces or torques at either end) we have

$$\begin{aligned} s = 0: \quad &\kappa\ddot\psi(0) = af(0) \;, \quad \tau(0) = 0 \;, \quad \kappa\dot\psi(0) = aF_b \\ s = L: \quad &\kappa\ddot\psi(L) = af(L) \;, \quad \tau(L) = 0 \;, \quad \dot\psi(L) = 0 \;, \end{aligned} \tag{20}$$

where the basal force is $F_b = k_b\Delta_b + \xi_b\partial_t\Delta_b$. *Equation 14–16* together with the boundary conditions and the force-response equation *Equation 6* constitute the full, non-linear model for the axonemal beat. Rather than solving this non-linear model, we only considered the first two modes and assumed that the static curvature was constant. This leads to an ordinary differential equation in arc-length for the tangent angle, with constant coefficients. A discrete set of solutions can then be obtained by solving the boundary value problem as described in *Appendix 2*. Each periodic solution corresponds to the critical point of a non-linear dynamic solution at the point of a Hopf bifurcation, as described in *Camalet et al. (1999)*.

## 2. Asymmetric equation for the fundamental mode

The periodic dynamics of the tangent angle can be decomposed into Fourier modes as defined in *Equation 2*. For asymmetric beating patterns in which $\dot\psi_0 \neq 0$, the static mode is characterized by the force balance $\kappa\dot\psi_0 = aF_0$, obtained from integrating *Equation 12* and using the boundary conditions. The static mode of the tension vanishes (also by *Equation 12*). The dynamics of a small amplitude oscillation dominated by the fundamental mode can be described by expanding *Equation 14 and 15* around the static component and keeping only terms in the $n=1$ mode. This results in

$$\begin{aligned} i\bar\omega\psi_1 &= -\dddot\psi_1 + \ddot f_1 + \dot\psi_0\dot\tau_1 + \ddot\psi_0\tau_1 + \xi\dot\psi_0(\kappa\dot\psi_0\ddot\psi_1 - \dot\psi_0 f_1 + \dot\tau_1), \\ \xi\ddot\tau_1 - \dot\psi_0^2\tau_1 &= -\dot\psi_0(\ddot\psi_1 - \dot f_1) + \xi\partial_s[\dot\psi_0(f_1 - \ddot\psi_1)], \end{aligned} \tag{21}$$

The equations above have been made dimensionless using the following rescalings: $\xi = \xi_n/\xi_t$, $\bar s = s/L$, $\bar\Delta = \Delta/a$, $\bar\omega = \xi_n\omega L^4/\kappa$, $\bar f = aL^2 f/\kappa$, $\bar\tau = L^2\tau/\kappa$, $\bar k_b = a^2 Lk_b/\kappa$ and $\bar\xi_b = a^2 L\omega\xi_b/\kappa$. This choice results in the additional rescalings $\bar f_\perp = aL^2 f_\perp/\kappa$, $\bar\chi = a^2 L^2\chi/\kappa$, $\bar\beta = aL\beta/\kappa$ and $\bar\gamma = \gamma$, since $\gamma$ is already dimensionless. This pair of equations is the generalization of the equations for the symmetric beat (*Machin, 1958*; *Camalet et al., 1999*). In them, the fundamental mode is coupled to the static mode. For the particular case in which the static shape has constant curvature $\dot\psi_0 = C_0$, *Equation 21* reduces to the asymmetric beat equations used in the main text (*Equation 5*).

*Equation 5* together with *Equation 6* form a system of ordinary differential equations with constant coefficients. Using the boundary conditions, the discrete spectrum of solutions can be obtained (*Cross and Hohenberg, 1993*; *Camalet and Jülicher, 2000*; *Sartori, 2015*)]. While the system is of sixth order, it contains an integral term in the expression for the normal force, *Equation 7*. It is thus convenient to convert the system to seventh order by taking the derivative of *Equation 7*, which eliminates the integral term. Provided values for the response coefficients $\chi$, $\beta$ and $\gamma$ we can then use the *ansatz* $\psi_1 = Ae^{ks}$ to obtain a characteristic

polynomial of order seven in $k$. The general solution to the boundary value problem is $\psi_1 = \sum_{i=1}^{7} A_i e^{k_i s}$, where the roots $k_i(\chi, \beta, \gamma)$ of the characteristic polynomial are implicit functions of the motor response coefficients. The amplitudes $A_i$ are then determined, up to an arbitrary factor, imposing that the boundary conditions be satisfied. Determining the amplitudes will in turn result in a fixed discrete spectrum of solutions for the possible basal compliances. Conversely, if the basal compliance is provided, calculating the amplitudes will return a discrete set of solutions for the real and imaginary parts of one of the response coefficients. These are the critical modes in *Camalet and Jülicher (2000)*, two examples are shown in *Figure 5*.

## 3. Fitting procedure

The fitting procedure was done as follows. Given a set of values for the response coefficients $\chi$, $\beta$ and $\gamma$, a theoretical solution $\psi_{\mathrm{the}}$ was obtained in the manner described in the previous section, up to an arbitrary complex amplitude. Given this solution, the force balance $\chi_{\mathrm{b}}\Delta_{\mathrm{b}} = F(0)$ allows us to determine $\chi_{\mathrm{b}}$. If the value for the real or imaginary parts of $\chi_{\mathrm{b}}$ were negative, corresponding to an active base, the solution was discarded. If they were positive, then the complex amplitude was chosen as to minimize the mean square displacement $R^2(\psi_{\mathrm{the}}, \psi_{\mathrm{exp}})$ given by

$$R^2 = 1 - \frac{\sum_{i=1}^{N} |\psi_{\mathrm{exp}}(s_i) - \psi_{\mathrm{the}}(s_i)|^2}{\sum_{i=1}^{N} |\psi_{\mathrm{exp}}(s_i)|} \quad , \tag{22}$$

where $N = 20$ and the points $s_i$ were equally spaced along the axonemal length. Finally, a value of $R^2 < 1$ was obtained. This quality function, which takes the response coefficients as an input, was maximized with the routine Find Minimum of Mathematica 10 using the Principal Axis method.

## 4. Sign convention

The signs of the geometric parameters such as sliding displacement, tangent angle and curvature, as well as the forces, are defined in *Figure 10*. The tangent angle is measured with respect to the horizontal x axis and grows counter-clockwise (the xy frame has the usual orientation). When a dynein motor is attached to the upper filament and walks towards the base on the lower filament (corresponding to the minus end of the microtubule), the filament pair bends downwards: the tangent angle is therefore negative ($\psi < 0$), as is the curvature ($\dot{\psi} < 0$). The sliding displacement, in which the lower filament extends beyond the upper filament, is therefore also negative ($\Delta < 0$). The force generated by this motor on the lower filament is defined as positive ($f > 0$) as the lower filament slides with its distal end leading. The integrated force acting at the base, $F_{\mathrm{b}} = -\int_0^L f(s) \mathrm{d}s$ is therefore negative. This is consistent with the static force balance equation, $\kappa\dot{\psi}(0) = aF_{\mathrm{b}}$, in which the basal curvature ($\dot{\psi}(0)$) and the basal force have the same sign. The sliding force will tend to make the upper filament extend leftwards beyond the lower one; because the basal force and the basal shear have the same sign, this shear is therefore negative ($\Delta_{\mathrm{b}} < 0$).

The signs of the parameters shown in the Figures and defined in the equations are consistent with this convention. For example, because the flagellum swims counterclockwise, the tangent angle slowly grows more positive over time, as shown in *Figure 2Aiii*. Also, the average angle shown in *Figure 3Aii* has a negative slope, which corresponds to a negative static curvature $C_0 < 0$.

## 5. Estimation of mechanical parameters of the axoneme

The only mechanical parameters entering the problem are the bending rigidity $\kappa$, the spacing $a$ of the filaments, and the two friction coefficients $\xi_n$ and $\xi_t$.

The bending rigidity of a single microtubule is $\approx 20 \, \mathrm{pN} \, \mu\mathrm{m}^2$ (**Gittes et al., 1993**) and see (**Schaedel et al., 2015**) and the bending rigidity of one doublet is expected to be approximately three times that of a single microtubule (**Howard, 2001**). Together, the axoneme is comprised of 9 doublet and 2 singlet microtubules which amounts to a bending rigidity of $580 \, \mathrm{pN} \, \mu\mathrm{m}^2$.

The diameter $D$ of the axoneme, determined from a circle through the centers of the 9 doublets, is $192 \pm 4$ nm (SD, n=19), measured from two different axonemes in **Figure 1** of **Bui et al. (2012)** and **Nicastro (2006)**. The spacing between the centers of adjacent doublets, $a$, is given by $a = D \sin(\pi/9)$, which is 66 nm. The hydrodynamic radius $r$ of the axoneme was calculated as $r = (D + 25 \, \mathrm{nm})/2$ and was 108 nm.

The tangential and normal friction coefficients per unit length of a slender rod of hydrodynamic radius $r$ near a surface (at height $h$ from surface to rod center) are given by $\xi_t \approx 2\pi\eta/ln(2h/r)$, where $\eta$ is the viscosity and $\xi_n = 2\xi_t$ (**Riedel-Kruse et al., 2007**). Considering the axoneme being immersed in water at $24°C$ ($\eta = 0.91 \, 10^{-3} \mathrm{pN} \, \mathrm{s} \, \mu\mathrm{m}^{-2}$), and beating at height $h = 0.55 \, \mu$m, from the surface, then $\xi_t \approx 0.0025 \, \mathrm{pN} \, \mathrm{s} \, \mu\mathrm{m}^{-2}$ and $\xi_n \approx 0.0049 \, \mathrm{pN} \, \mathrm{s} \, \mu\mathrm{m}^{-2}$.

