## [Decision Letter]

Thank you for submitting your work entitled "Dynamic curvature regulation accounts for the symmetric and asymmetric beats of *Chlamydomonas* flagella" for consideration by *eLife*. Your article has been reviewed by two peer reviewers, and the evaluation has been overseen by Raymond E. Goldstein as the Reviewing Editor and Aviv Regev as the Senior Editor. One of the reviewers has agreed to reveal his identity: Michael J. Shelley.

The reviewers have discussed the reviews with one another and the Reviewing Editor has drafted this decision to help you prepare a revised submission.

Summary:

Through a combination of experiment and theory this paper analyzes the beating waveforms of *Chlamydomonas* flagella as a means of distinguishing between different microscopic models. Using isolated reactivated axonemes of both wildtype and mutant *Chlamydomonas* the authors find strong evidence that regulation by curvature if favoured over that by sliding or normal forces. In addition, the analysis supports the regulation by the time-varying curvature rather than its static value.

Essential revisions:

1) The reviewers were in agreement that the paper, while important and insightful, is difficult to follow in its present form. For instance, it is not always very well organized, especially since the fully nonlinear model is never fully specified, and one has to bounce around the text to try and piece together what is being done.

2) Accessibility. The mathematics behind the continuum model is not easily accessible to a general biological readership. Although the authors have done some work to make it more easily understandable, we feel that more needs to be done. One of the main things is to extend the level of explicit derivation within the Appendix, and then in the main paper reference to specific equations, rather than generically to "the Appendix". Also, it would be beneficial to expand on points like: i) why does (4) show that an oscillatory *f_1_* produces dynamic bending? ii) in the subsection second paragraph of the subsection “Three mechanisms of motor control”, why are sliding and curvature controls "linear" (in what?) and normal force control "non-linear"? iii) in the subsection “Dynamic curvature control as a mechanism for motor regulation”, why does the dependence on time-derivative of the curvature follow from *β(ω)* having no real part? iv) in the Appendix, make more explicit the interpretation of the boundary conditions.

3) With respect to that model itself, it would be nice to see a better explanation of how two coupled elastica moving under the constraints of equal spacing and (equal and opposite) sliding interactions, must yield the form of center-line dynamics given in (11). Also, has there been any fuller 1st principles modeling, say from microscopics, of the form that sliding interactions must take, especially under active forcing from motor-proteins ala Figure 1?

4) Please clarify that the nonlinear simulations were performed using models for sliding regulation – at least by sliding or curvature feedback – using constant coefficient functions? If not, what were they (that is, what were their frequency dependencies). Was there an s-dependence in those coefficient functions?

5) Having made the arguments against sliding and normal force mechanisms and for curvature control, what would be the prediction of this model for changes in beat frequency and shape for increases in solvent viscosity? (ala Figure 4 in Fauci's Annual Reviews of Fluid Mechanics article).

6) The wild type time-averaged shape is claimed to have constant curvature, equivalent to the black curve in Figure 3(ii) being a line. We are not convinced that this is the case. At the very least there is an initial bend, then again a change in slope at ~5μm. We would like to see a plot of the first and second derivatives of *ψ_0_(s),* showing directly the local curvature and the local force component *f_0_(s)*. Looking at the curve, it seems that there will be components to *f_0_(s)* localised around the base and at *s*~5μm. How would these affect the conclusions on the *n*=0 behaviour of the motor response coefficients?

7) The authors claim that the force leading to the constant curvature is localised at the tip. However, there are no molecular motors at the flagellar tip. Can the authors comment on this discrepancy?

8) One of the main conclusions of the paper is the suggestion that the static and dynamic components of the curvature are regulated independently. The authors comment on the origin of the dynamic regulation. Could they also comment on the origin of the static curvature? Based on what is known of the *mbo2* mutation, is it possible to speculate on how might this affect the static curvature?

9) The authors state that the 2D model oversimplifies the effect of normal force, and have developed a 3D model within a working paper (Sartori et al., 2015) which from its title suggest that normal forces *can* account for motor regulation. Why, then, was the 2D model used here instead of the 3D one? Wouldn't the conclusions presented here be negated, instead of confirmed, by the full 3D model?

10) In the variational principle outlined in the Appendix the contribution to the energy functional from the work done by the motors is written as *fΔ* and the variation is done assuming only that *f* is a function of arclength but not itself functionally dependent on the angle *ψ*. But later when specific forms of *f* are considered they do depend on the tangent angle *ψ*. This appears to be an inconsistency and we request clarification.

---

## [Author Response]

*Essential revisions:*

*1) The reviewers were in agreement that the paper, while important and insightful, is difficult to follow in its present form. For instance, it is not always very well organized, especially since the fully nonlinear model is never fully specified, and one has to bounce around the text to try and piece together what is being done.*

We apologize that the manuscript was difficult to follow. We have made many revisions in the main text, the Theory and the Appendix and we hope that the overall argument, including the mathematical arguments, are now much clearer. Regarding the non-linear model, we actually solve a linearized model, and now make that very clear in the revised text.

*2) Accessibility. The mathematics behind the continuum model is not easily accessible to a general biological readership. Although the authors have done some work to make it more easily understandable, we feel that more needs to be done. One of the main things is to extend the level of explicit derivation within the Appendix, and then in the main paper reference to specific equations, rather than generically to "the Appendix". Also, it would be beneficial to expand on points like: i) why does (4) show that an oscillatory f_1_ produces dynamic bending? ii) in the subsection second paragraph of the subsection “Three mechanisms of motor control”, why are sliding and curvature controls "linear" (in what?) and normal force control "non-linear"? iii) in the subsection “Dynamic curvature control as a mechanism for motor regulation”, why does the dependence on time-derivative of the curvature follow from β(ω) having no real part? iv) in the Appendix, make more explicit the interpretation of the boundary conditions.*

We have put a lot of effort into making the paper accessible for a general biological readership.

A) We have substantially rewritten the Introduction to get across the main ideas of reciprocal inhibition of motors across the section of the axoneme.

B) In the Theory, we have added extra panels in Figure 1 to show how the 2D model is derived from the 3D model and how the dyneins on opposite sides of the axoneme lead to bending in opposite directions.

C) We have also made the predictions of the dynamic curvature model clear in Figure 9, showing precisely where and when the motors are active relative to the curvature of the axoneme. These predictions can then be directly tested by structural studies, such as cryoEM.

D) We have made the mathematical analysis in the Appendix more logical and have referenced the specific equations in the main paper.

E)We have also expanded on the four points mentioned, as well as many others:

i) We explain explicitly how an oscillatory force gives an oscillating waveform;

ii) Why sliding and curvature control are linear and normal-force control not (though it becomes linear in the presence of constant static curvature);

iii) The significance of the imaginary part of the control coefficients;

iv) How the boundary conditions arise (i.e. no external forces or torques).

*3) With respect to that model itself, it would be nice to see a better explanation of how two coupled elastica moving under the constraints of equal spacing and (equal and opposite) sliding interactions, must yield the form of center-line dynamics given in (11). Also, has there been any fuller 1st principles modeling, say from microscopics, of the form that sliding interactions must take, especially under active forcing from motor-proteins ala Figure 1?*

We provide references to earlier papers (Camalet & Jülicher 2000 and Mukundan et al. 2014) for a more detailed treatment of the equal spacing constraint. We reference a paper from the Bayly group at U. Washington that solved a microscopic model; however, we feel that the ODE-based approach here is well suited for fitting to experimental data because there is a clear link between the parameters and the shape.

*4) Please clarify that the nonlinear simulations were performed using models for sliding regulation – at least by sliding or curvature feedback – using constant coefficient functions? If not, what were they (that is, what were their frequency dependencies). Was there an s-dependence in those coefficient functions?*

First, we linearized the models so that they are ODEs with constant coefficients. We solved the associated boundary value problem to identify the critical point solutions. The coefficients are all assumed to be independent of arc length, and we have made this clear. The coefficients depend on frequency, but because we only consider the first mode, the curve fitting yields parameters at that frequency only.

*5) Having made the arguments against sliding and normal force mechanisms and for curvature control, what would be the prediction of this model for changes in beat frequency and shape for increases in solvent viscosity? (ala Figure 4 in Fauci's Annual Reviews of Fluid Mechanics article).*

This is a very good and important question. This is still work in progress. Note that the effects of viscosity in the sliding- control model were computed in Camalet and Jülicher (2000).

*6) The wild type time-averaged shape is claimed to have constant curvature, equivalent to the black curve in Figure 3(ii) being a line. We are not convinced that this is the case. At the very least there is an initial bend, then again a change in slope at ~5μm. We would like to see a plot of the first and second derivatives of ψ_0_(s), showing directly the local curvature and the local force component f_0_(s). Looking at the curve, it seems that there will be components to f_0_(s) localised around the base and at s~5μm. How would these affect the conclusions on the n=0 behaviour of the motor response coefficients?*

This is a reasonable question. Figure 11 shows the requested plots. The reviewers are correct that the static curvature is not constant. We have qualified our statements in the text to say that the curvature is only approximately constant.

Author response image 1.**DOI:**
http://dx.doi.org/10.7554/eLife.13258.014

The curvature and the basal and distal ends approaches zero. It is difficult to measure curvature at the ends and we were worried about potential artifacts; we are confident, however that the curvature decreases at both ends. At the distal end it is easy to understand: there are no external torques. See the Mukundan et al. 2014 paper for a possible model. The decrease in the curvature at the basal end is presumably due to the (static) compliance at the base. Importantly, the curvature is fairly constant over the central region of the axoneme: the standard deviation is about 20% of the mean. We therefore model the curvature as being due to sliding motors active near (but not at the distal end). Note that an alternative model in which motors are active all along the length would give a static shape that is a spiral (see Mukundan et al. 2014), which is certainly not the case.

One reason we think that the constant curvature approximation is a good one, is that the waveform data shows that the dynamic beats are similar in the wild type and the *mbo2* mutant, so the large static curvature cannot be critical to the beat. Furthermore, in the curvature-control model, the effect of static curvature is small.

*7) The authors claim that the force leading to the constant curvature is localised at the tip. However, there are no molecular motors at the flagellar tip. Can the authors comment on this discrepancy?*

We have qualified this statement by saying the force is mainly (but not completely) localized near (but not at) the tip.

*8) One of the main conclusions of the paper is the suggestion that the static and dynamic components of the curvature are regulated independently. The authors comment on the origin of the dynamic regulation. Could they also comment on the origin of the static curvature? Based on what is known of the mbo2 mutation, is it possible to speculate on how might this affect the static curvature?*

It is very difficult to interpret the *mbo2* mutant. Part of the problem is that most structural mutants in *Chlamydomonas* (i.e. mutants that are observed by EM to be missing structural elements such as dyneins and regulatory complexes) are *assembly* mutants, meaning that they miss proteins required to assemble the structures but do not necessarily encode for the missing proteins.

*9) The authors state that the 2D model oversimplifies the effect of normal force, and have developed a 3D model within a working paper (Sartori et al., 2015) which from its title suggest that normal forces can account for motor regulation. Why, then, was the 2D model used here instead of the 3D one? Wouldn't the conclusions presented here be negated, instead of confirmed, by the full 3D model?*

We have uploaded the 3D paper to the arXiv so that it is accessible to the reviewers. In this paper, we show that if the doublets twist around the surface of the axoneme, then there is coupling between curvature and transverse force (which in the 2D model becomes a normal force that separates the two filaments). This is of course a hypothesis that would need to be tested. However, in the absence of twist, the transverse force is symmetric as is the normal force in the 2D model (i.e. the transverse force is expansive on both sides of the axoneme and so cannot lead to reciprocal inhibition). In other words, our 2D model captures the behavior of an untwisted 3D axoneme. So the 3D model does support the present conclusions.

*10) In the variational principle outlined in the Appendix the contribution to the energy functional from the work done by the motors is written as fΔ and the variation is done assuming only that f is a function of arclength but not itself functionally dependent on the angle ψ. But later when specific forms of f are considered they do depend on the tangent angle ψ. This appears to be an inconsistency and we request clarification.*

We have clarified this point in the text.

We need to know the instantaneous external force *f*_ext_ (that acts on the fluid), given the shape and the internal stresses due to the force f between filaments (generated by the motors). This relation between *f*_ext_ and *f* for a given shape corresponds to an instantaneous force balance in the filament.

From this we can then write the dynamic equations by also balancing *f*_ext_ with viscous forces in the fluid.

To derive the instantaneous force *f*_ext_, we define the work *dW*=*f*_ext_*dr* exerted by the external forces under a small displacement. If we keep *f* constant during such a displacement (no change in *f(s)* while the displacement is being changed), then the work *dW*=*dU* can be written as a change of a potential, where *U* is the potential defined in (11) in the appendix. The external force *f*_ext_=*δU/δr* can thus be calculated from (11) by variation using *f(s*)=const.

If *f(s*) also varies by *δf(s*) when the displacement *δr* occurs, then *dW* can in general not be written as a change *dU* of a potential. The force balance condition between *f*_ext_ and *f* always remains the same but *dW*=*dU* is valid only for constant *f*. For a dynamic flagellum with time varying motor forces *f*, the forces *f*_ext_ are not potential forces because this is a nonequilibrium system. However, *f*_ext_ still obeys the instantaneous force balance obtained for *δf*=0 from the potential *U*.